# Confidence-Aware With Prototype Alignment for Partial Multi-label Learning

**Weijun Lv**[1], **Yu Chen**[1], **Xiaozhao Fang**[1]*, **Xuhuan Zhu**[1], **Jie Wen**[2]*,
**Guoxu Zhou**[1], **Sixian Chan**[3]

[1]School of Automation, Guangdong University of Technology, Guangzhou 510006, China
[2]Shenzhen Key Laboratory of Visual Object Detection and Recognition,
Harbin Institute of Technology (Shenzhen), Shenzhen 518055, China
[3]College of Computer Science and Technology, Zhejiang University of Technology,
Hangzhou 310023, China
{lvweijun0201, chenyu9265324}@163.com, {xzhfang168, xuhuanz_zz}@126.com
jiewen_pr@126.com, gx.zhou@gdut.edu.cn, sxchan@zjut.edu.cn

## Abstract

Label prototype learning has emerged as an effective paradigm in Partial Multi-Label Learning (PML), providing a distinctive framework for modeling structured representations of label semantics while naturally filtering noise through prototype-based label confidence estimation. However, existing prototype-based methods face a critical limitation: class prototypes are the biased estimates due to noisy candidate labels, particularly when positive samples are scarce. To this end, we first propose a mutually class prototype alignment strategy bypassing noise interference by introducing two different transformation matrices, which makes the class prototypes learned by the fuzzy clustering and candidate label set mutually alignment for correcting themselves. Such alignment is also passed on to the fuzzy memberships label in turn. In addition, to eliminate noise interference in the candidate label set during the classifier learning, we use the learned permutation matrix to transform the fuzzy memberships label for learning a label reliability indicator matrix accompanied by the candidate label set. This makes the label reliability indicator matrix absolutely prevent the occurrence of numerical values located in non-label and simultaneously eliminate the introduction of incorrect label as much as possible. The resulting indicator matrix guides a robust multi-label classifier training process, jointly optimizing label confidence and classifier parameters. Extensive experiments demonstrate that our proposed model exhibits significant performance advantages over state-of-the-art PML approaches.

## 1 Introduction

Multi-Label Learning (MLL) is an important branch of machine learning that allows an instance to belong to multiple categories simultaneously, with widespread applications in image annotation[1, 2], text categorization[3], and medical diagnosis [4]. However, in practical application scenarios, obtaining precisely annotated data is often difficult due to high annotation costs and inherent ambiguity[5, 6, 7, 8]. Annotators frequently provide a candidate label set when uncertain, containing both ground-truth labels and noisy labels (i.e., false positive labels mistakenly included). For example, in image annotation tasks, objects with similar visual appearances may be incorrectly labeled, which will introduce interference to the classification prediction. To address this issue, Partial Multi-Label Learning (PML) [9] has emerged as a new weakly supervised learning framework. The

---

*Corresponding Author

39th Conference on Neural Information Processing Systems (NeurIPS 2025).

main task of PML is to learn from this uncertain supervisory information to accurately predict the ground-truth labels for unknown instances.

PML research primarily focuses on identifying ground-truth labels from the candidate label set, known as label disambiguation. Existing methods mostly adopt explicit learning strategies, directly removing noisy labels from candidate label sets through various techniques. For instance, some methods use low-rank and sparse decomposition to separate ground-truth labels from noisy ones [10, 11]; others estimate the credibility of candidate labels through label propagation or label distribution learning [12, 13, 14, 15]; while some utilize feature information to identify noisy labels [16, 5], where the practice of using feature information to impose correlation constraints or manifold constraints on labels is the most popular [17, 18, 19]. Recently, feature selection has become a popular approach for disambiguation [20, 21, 22]. Some researchers leverage label correlations or cluster assignments to assist disambiguation [9, 23, 24, 25, 26, 19]. Additionally, some methods introduce complementary classifiers that simultaneously leverage both positive and negative label information under the noise sparsity assumption [27]. Another line of research adopts a phased processing approach: first purifying noisy labels through methods like granular ball construction [25] or label propagation [12], then applying structured learning strategies such as confidence score regression or pairwise classification paradigms [28, 29] to the refined labels. These explicit learning approaches, although effective to some extent, still face limitations when handling complex noise patterns. On one hand, methods based on sparsity assumptions often fail when real-world annotation scenarios produce non-sparse noise distributions, leading to error accumulation during classifier training. On the other hand, methods based on label confidence or manifold learning struggle with samples lacking sufficient neighborhood information or when positive samples are scarce, making it difficult to fully fit the data distribution.

Recently, prototype-based learning[9, 25, 30, 19] has emerged as a promising direction that constructs class prototypes to guide label confidence estimation. As a cornerstone of prototype-based approaches, clustering learning serves as an important tool for exploratory data analysis [31, 32], naturally generating class prototypes without requiring prior labels while efficiently revealing inherent data structure [33, 34]. Building upon fuzzy clustering principles, FBD-PML [19] advances this direction by mining the correlation between sample instances and labels while learning confidence values under sample manifold assumptions. However, existing prototype-based methods face a critical limitation: prototypes derived from noisy candidate labels inevitably deviate from true semantic centers—a representation bias problem pervasive under imperfect supervision[35, 36, 37, 38]. This fundamental weakness becomes particularly pronounced when positive samples are scarce, as the limited reliable supervision further exacerbates prototype distortion. This prototype bias propagates through confidence estimation, ultimately degrading classification performance. Moreover, existing methods struggle to effectively bridge the semantic gap between unsupervised clustering-derived prototypes and weakly supervised label-based prototypes, leaving the potential complementarity between these two prototype spaces largely unexplored.

Addressing these challenges, this paper proposes CAPML (**C**onfidence-**A**ware with Prototype Matching for **P**artial **M**ulti-label **L**earning), a novel method that leverages unsupervised clustering to bypass noise interference while enhancing weakly-supervised semantic representations through effective prototype space alignment. Specifically, unlike traditional prototype-based methods that solely rely on noisy candidate labels, the proposed approach introduces a transformation mechanism that successfully bridges the gap between clustering-derived prototypes and label-based prototypes, enabling discovery of their intrinsic correspondence despite noisy supervision. Subsequently, a confidence-aware process is designed to convert fuzzy membership degrees into label reliability indicators, guiding classifier training with sparse $\ell_{2,1}$-norm constraints. Concurrently, the prototype alignment mechanism is also utilized to guide the refinement of label confidence estimation. Finally, the enhanced confidence values and learned classifiers work jointly to predict labels for unknown instances. The main contributions of this paper are summarized as follows:

- This paper provides the first investigation into prototype misalignment between prototypes derived from fuzzy clustering and prototypes computed from candidate label sets in PML tasks, introducing a transformation mechanism that successfully bridges these two prototype spaces and enables effective alignment despite noisy supervision.
- This paper designs a confidence-aware process that converts fuzzy label membership degrees into label reliability indicator values, guiding classifier training with sparse $\ell_{2,1}$-norm constraints that enhance feature selection while reducing overfitting to noisy labels.

- Extensive empirical evaluation demonstrates the proposed method's efficacy in resolving label ambiguity and prototype misalignment problems in PML, achieving superior generalization performance even with high noise rates and sparse positive samples.

This section introduces the concept of PML, its evolution, related work, and how our approach differs from existing methods. The remainder of this paper is organized as follows: Section 2 details the principles and optimization algorithm of CAPML. Section 3 and 4 provide comprehensive experimental results and analysis. Finally, Section 5 gives a conclusion and Section 6 illustrates the limitations of the method proposed.

## 2 Method

In PML, let $\mathcal{X} \subset \mathbb{R}^d$ denotes the $d$-dimensional feature space and $\mathcal{Y} = \{\boldsymbol{l}_1, \boldsymbol{l}_2, ..., \boldsymbol{l}_q\}$ denotes the label space with $q$ class labels. The training dataset $\mathcal{D} = \{(\boldsymbol{x}_i, \boldsymbol{y}_i)|1 \leq i \leq n\}$ contains $n$ examples, where $\boldsymbol{x}_i \in \mathcal{X}$ is the $i$-th instance and $\boldsymbol{y}_i \in \{0, 1\}^q$ represents its corresponding label vector. Let $\mathbf{X} = [\boldsymbol{x}_1, \boldsymbol{x}_2, \ldots, \boldsymbol{x}_n] \in \mathbb{R}^{d \times n}$ denotes the feature matrix and $\mathbf{Y} = [\boldsymbol{y}_1, \boldsymbol{y}_2, \ldots, \boldsymbol{y}_n]^\top \in \{0, 1\}^{n \times q}$ represents the candidate label matrix containing wrong annotation. Here, $y_{ij} = 1$ indicates the $i$-th instance is annotated with the $j$-th label, and $y_{ij} = 0$ indicates otherwise. Each instance correspond to a set of candidate labels with unrelated labels incorrectly labeled as 1, which is called noisy label. The goal of PML is to learn a classification function $f : \mathcal{X} \rightarrow 2^{\mathcal{Y}}$ that minimizes the effect of noisy label information and makes accurate label predictions. Figure .1 (right) illustrates the overall architecture of CAPML.

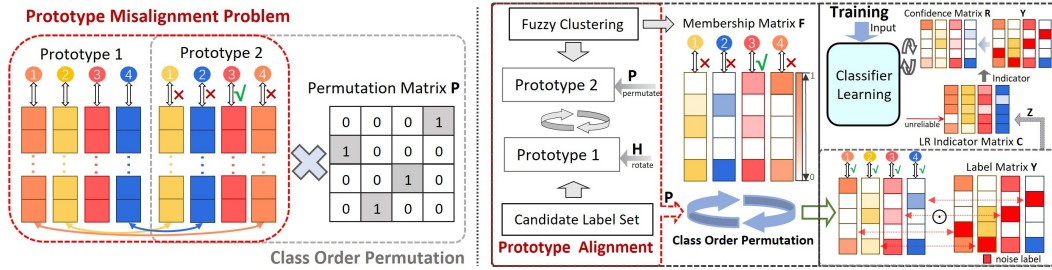

Figure 1: Overview of the CAPML framework.
**Left**: The prototype misalignment problem—unsupervised prototypes (Prototype 1) and weakly supervised prototypes (Prototype 2) lack optimal correspondence that require permutation matrix $\mathbf{P}$ for class order alignment.
**Right**: The two-stage pipeline: (1) *Prototype learning and alignment*—fuzzy clustering produces membership matrix $\mathbf{F}$ and unsupervised Prototype 1 while weakly supervised Prototype 2 is derived from candidate label set, then permutation matrix $\mathbf{P}$ establishes optimal correspondence, transforming memberships into label reliability (LR) indicator matrix $\mathbf{C}$ via element-wise product with $\mathbf{Y}$ and normalization; (2) *Confidence-aware classifier training*—indicator matrix $\mathbf{C}$ guides the learning of label confidence matrix $\mathbf{R}$, which in turn supervises classifier training to effectively suppress noise labels (marked in red) in $\mathbf{Y}$ and obtain proper predictions.

### 2.1 Class Prototype Learning and Alignment

Our approach begins with unsupervised prototype learning to capture the intrinsic structure of the data, independent of potentially noisy label information. We employ an improved fuzzy clustering approach incorporating entropy regularization[39, 40] to obtain well-distributed class prototypes and reliable fuzzy label membership degrees. The unified objective function is formulated as:

$$\min_{\mathbf{F}\mathbf{1}_c=\mathbf{1}_n, \mathbf{F} \geq 0, \mathbf{M}} \sum_{i=1}^{n}\sum_{j=1}^{c} f_{ij}\|\boldsymbol{x}_i - \boldsymbol{m}_j\|_2^2 + \lambda \sum_{i=1}^{n}\sum_{j=1}^{c} f_{ij} \log f_{ij}, \tag{1}$$

where $\mathbf{M} \in \mathbb{R}^{d \times c}$ denotes the class prototype matrix, $\mathbf{F} \in \mathbb{R}^{n \times c}$ represents the fuzzy label membership matrix indicating the association strength between instances and class prototypes, and $\lambda$ controls the entropy regularization. The first term quantifies weighted clustering quality, while

the entropy term prevents degenerate solutions and enables flexible membership distributions that can accommodate the multi-label nature by smoothly transitioning from unimodal to multimodal assignments. Through alternating optimization, the update formulas for $\mathbf{M}$ and $\mathbf{F}$ as follows:

$$\boldsymbol{m}_j = \frac{\sum_{k=1}^n f_{kj} \boldsymbol{x}_k}{\sum_{k=1}^n f_{kj}}, \ f_{ij} = \frac{e^{-\frac{\|\boldsymbol{x}_i - \boldsymbol{m}_j\|_2^2}{\lambda}}}{\sum_{k=1}^c e^{-\frac{\|\boldsymbol{x}_i - \boldsymbol{m}_k\|_2^2}{\lambda}}}. \tag{2}$$

The membership degree $f_{ij}$ follows a softmax-like formulation, where instances have higher membership degrees to closer prototypes. The parameter $\lambda$ controls the "softness" of the assignments—larger values result in more uniform distributions, while smaller values lead to more decisive assignments. While the prototypes M derived from fuzzy clustering lack explicit semantic meaning, we can leverage them by setting $c = q$ to match the number of label classes. To incorporate label semantic information, we construct class prototypes $\mathbf{O} \in \mathbb{R}^{d \times q}$ from the candidate label matrix $\mathbf{Y}$:

$$\boldsymbol{o}_j = \frac{\sum_{k=1}^n y_{kj} \boldsymbol{x}_k}{\sum_{k=1}^n y_{kj}}. \tag{3}$$

Each prototype $\boldsymbol{o}_j$ is computed as the centroid of instances associated with the $j$-th label in $\mathbf{Y}$. Despite the presence of noise, $\mathbf{O}$ can be viewed as weakly-supervised prototype[41] that still contains valuable semantic information. Although $\mathbf{M}$ and $\mathbf{O}$ are derived from different principles—unsupervised clustering and weakly supervised label aggregation respectively—they fundamentally capture the same underlying class structure.

Then a critical problem emerges: *although both* $\mathbf{M}$ *and* $\mathbf{O}$ *capture class-level representation, their orderings are inherently misaligned due to the arbitrary indexing produced by clustering.* For example, as shown in Figure .1 (left), only the 3-rd prototype in $\mathbf{M}$ correctly corresponds to its counterpart in $\mathbf{O}$, while the remaining 1-st, 2-nd, 4-th unsupervised prototypes improperly correspond to positions 4-th, 1-st, 2-nd respectively. To address this issue, we introduce a permutation matrix $\mathbf{P} \in \{0, 1\}^{q \times q}$ to align these orders:

$$\min_{\mathbf{P}} \|\mathbf{MP} - \mathbf{O}\|_F^2, \quad s.t. \ \mathbf{P1}_q = \mathbf{1}_q, \ \mathbf{P}^\top \mathbf{1}_q = \mathbf{1}_q, \ \mathbf{P} \in \{\mathbf{0}, \mathbf{1}\}^q. \tag{4}$$

This formulation seeks the optimal one-to-one mapping between unsupervised and weakly supervised prototypes by minimizing the Frobenius norm of their difference. The constraints ensure that $\mathbf{P}$ is a valid permutation matrix, with exactly one entry of 1 in each row and column.

**Theorem 1**: *When the fuzzy clustering successfully captures the underlying class structure, there exists an optimal permutation matrix* $\mathbf{P}^* \in \{0, 1\}^{q \times q}$ *such that the alignment error satisfies:*

$$\|\mathbf{MP}^* - \mathbf{O}\|_F = O\left(\epsilon \sqrt{q} + \sqrt{\frac{q^2}{n}}\right), \tag{5}$$

*where* $\epsilon$ *measures the label noise level (specifically, the fraction of noisy labels in* $\mathbf{Y}$*),* $n$ *is the number of training instances, and* $q$ *is the number of classes.*

The proof is provided in Appendix. This theorem reveals that the alignment error is governed by two factors: (1) the label noise level $\epsilon$ in the candidate set, and (2) the finite sample effect $\sqrt{q^2/n}$, which diminishes as more training data becomes available. Importantly, the bound suggests that even with moderate noise, the permutation matrix $\mathbf{P}$ can establish meaningful correspondence when sufficient data is present.

However, another key challenge remains: *since* $\mathbf{Y}$ *contains noise, the weakly supervised prototypes* $\mathbf{O}$ *deviate from true class centroids.* Directly aligning $\mathbf{M}$ to $\mathbf{O}$ might propagate these noise-induced shifts. To address this discrepancy and make $\mathbf{P}$ more reliable, we introduce an orthogonal rotation matrix $\mathbf{H} \in \mathbb{R}^{q \times q}$ to allow for more flexible alignment:

$$\min_{\mathbf{P}, \mathbf{H}} \|\mathbf{MP} - \mathbf{OH}\|_F^2, \quad s.t. \ \mathbf{P1}_q = \mathbf{1}_q, \ \mathbf{P}^\top \mathbf{1}_q = \mathbf{1}_q, \ \mathbf{P} \in \{\mathbf{0}, \mathbf{1}\}^q, \ \mathbf{HH}^\top = \mathbf{I}_q. \tag{6}$$

The orthogonal rotation matrix $\mathbf{H}$ introduces geometric transformation that adapts the weakly supervised prototypes $\mathbf{O}$ to better match the unsupervised structure, while preserving their relative geometry through the constraint $\mathbf{HH}^\top = \mathbf{I}_q$. This mitigates the negative impact of label noise: rather than forcing $\mathbf{M}$ to directly align with the potentially biased $\mathbf{O}$, we allow $\mathbf{O}$ to rotate in its representation space, reducing the negative influence of noise-induced biases on the alignment quality.

**Constructing label reliability indicator.** Having obtained the optimal permutation matrix $\mathbf{P}$ and rotation matrix $\mathbf{H}$, we establish the correspondence between $\mathbf{M}$ and $\mathbf{O}$. The permutation matrix $\mathbf{P}$ not only aligns the prototype spaces but also reveals which cluster corresponds to which label class, thereby enabling transformation of the fuzzy membership degree in $\mathbf{F}$ into label-specific reliability indicator for candidate labels.

We apply the learned permutation matrix $\mathbf{P}$ to reorder the membership matrix $\mathbf{F}$, aligning each column with its corresponding label class. To mitigate interference from misaligned entries, we perform element-wise multiplication with the candidate label matrix $\mathbf{Y}$ to retain only candidate positions, followed by row-wise min-max normalization $\mathcal{N}_{\text{minmax}}(\cdot)$ to amplify confidence contrasts:

$$\mathbf{Z} = \mathcal{N}_{\text{minmax}}(\mathbf{F}\mathbf{P} \odot \mathbf{Y}). \tag{7}$$

This operation filters out non-candidate noise while enhancing discrimination between reliable and unreliable candidates. We then construct the label reliability (LR) indicator matrix $\mathbf{C} \in \mathbb{R}^{n \times q}$ as:

$$c_{ij} = \begin{cases} 1, & \text{if } y_{ij} = 0 \\ z_{ij}, & \text{if } y_{ij} = 1 \end{cases}. \tag{8}$$

However, this initial formulation treats all instances uniformly, ignoring that instances with more candidate labels typically contain more noise. To address this, we introduce instance-adaptive weighting based on candidate label density:

$$c_{ij} = \begin{cases} \|\mathbf{y}_i\|_1, & \text{if } y_{ij} = 0 \\ (q - \|\mathbf{y}_i\|_1) * z_{ij}, & \text{if } y_{ij} = 1 \end{cases}. \tag{9}$$

This refined formulation assigns differentiated indicator values based on candidate label density, which ensures that instances with denser candidate sets—which statistically contain more false positives—receive more conservative reliability estimates.

## 2.2 Confidence-Aware Label Disambiguation

The LR indicator matrix $\mathbf{C}$ plays a crucial role in our label disambiguation process. A higher indicator value $c_{ij}$ indicates a higher probability that $r_{ij}$ is a true label rather than noise. With the label reliability (LR) indicator matrix $\mathbf{C}$ constructed, we now formulate the confidence-aware objective for joint classifier learning and label disambiguation::

$$\min_{\mathbf{W},\mathbf{R}} \|\mathbf{C} \odot (\mathbf{Y} - \mathbf{R})\|_F^2 + \|\psi(\mathbf{X})^\top \mathbf{W} - \mathbf{R}\|_F^2 + \alpha\|\mathbf{Y} - \mathbf{R}\|_1 + \beta\|\mathbf{W}\|_{2,1}, \ s.t. \ \mathbf{R} \geq 0, \tag{10}$$

where $\psi(\cdot) : \mathbb{R}^d \to \mathbb{R}^h$ is a feature mapping function that transforms input features to a kernel space for better separability, $\mathbf{W} \in \mathbb{R}^{h \times q}$ represents the classifier parameters, $\mathbf{R} \in \mathbb{R}^{n \times q}$ denotes the label confidence matrix, and $\alpha, \beta$ are regularization hyperparameters. The first term implements reliability-weighted refinement through element-wise multiplication. For non-candidates, the weight $c_{ij} = 1$ penalizes any deviation from zero in $\mathbf{R}$, constraining $r_{ij} \approx 0$. For candidates, varying weights create differentiated penalties: high-reliability positions (large $c_{ij}$) are tightly constrained to $\mathbf{Y}$, while low-reliability positions (small $c_{ij}$) receive weaker constraints, allowing the classifier consistency and $\ell_1$ terms to guide their refinement. The $\ell_1$ norm term further reduces the influence of noise in candidate label set $\mathbf{Y}$, while the $\ell_{2,1}$ norm term enhances the classifier's discriminative power by emphasizing features with high discriminative capacity across multiple labels.

## 2.3 Optimization

The proposed approach optimization is divided into two parts. After obtaining $\mathbf{O}$ and $\mathbf{F}$ through Eq(2), we need to learn to obtain an effective permutation matrix $\mathbf{P}$ to obtain LE indicator matrix $\mathbf{C}$. Then $\mathbf{C}$ is reused for further refinement of the label confidence matrix $\mathbf{R}$ to guide the learning of the classifier $\mathbf{W}$. We optimize each variable by adopting an alternating iterative way.

**Update H, fix P.** We can obtain the following optimization problem about variant $\mathbf{H}$.

$$\min_{\mathbf{H}} \|\mathbf{A} - \mathbf{O}\mathbf{H}\|_F^2, \quad s.t. \ \mathbf{H}\mathbf{H}^\top = \mathbf{I}_q, \tag{11}$$

where $\mathbf{A} = \mathbf{M}\mathbf{P}$. Eq. (11) presents a standard orthogonal Procrustes problem [42], which we efficiently solve following the optimization approach detailed in [43].

**Algorithm 1:** Training Process of CAPML

---

**Input:** The PML training dataset $\mathcal{D}$; parameters $\lambda$, $\alpha$, $\beta$; max iterations $T_0$, $T_1$, $T_2$; Unseen sample $\hat{\mathbf{x}}$.
**Output:** the predicted label for unseen sample $\hat{\mathbf{y}}$.
// Stage One: Prototype Learning and Alignment
Initialize membership matrix $\mathbf{F} = \mathbf{Y}$ and prototypes $\mathbf{M}$ via Eq. (2). **for** $t = 1$ *to* $T_0$ **do**
   |    Update $\mathbf{M}$ and $\mathbf{F}$ via Eq. (2);        // Fuzzy clustering for unsupervised prototype
**end**
Compute $\mathbf{O}$ via Eq. (3);        // Compute supervised prototype from candidate labels
Initialize permutation matrix $\mathbf{P} = \mathbf{I}_q$ and rotation matrix $\mathbf{H} = \mathbf{I}_q$.
**for** $t = 1$ *to* $T_1$ **do**
   |    Update $\mathbf{H}$ by solving orthogonal Procrustes problem in Eq. (11)
   |    Update $\mathbf{P}$ using Hungarian algorithm in Eq. (12);     // Prototype alignment optimization
**end**
Compute label reliability indicator matrix $\mathbf{C}$ via Eq. (9) ;      // Construct LR indicator matrix
// Stage Two: Classifier Learning
Initialize $\mathbf{W} = \mathbf{0}_q$, $\mathbf{R} = \mathbf{Y}$, and auxiliary variable $\mathbf{Q} = \mathbf{0}_{n \times q}$.
**for** $t = 1$ *to* $T_2$ **do**
   |    Update $\mathbf{W}$ using Eq. (13)
   |    Update $\mathbf{R}$ using multiplicative rule in Eq. (16)
   |    Update $\mathbf{Q}$ using soft-thresholding via Eq. (19)
   |    **if** $\|\mathbf{W}^{(t)} - \mathbf{W}^{(t-1)}\|_F^2 + \|\mathbf{R}^{(t)} - \mathbf{R}^{(t-1)}\|_F^2 < 10^{-5}$ **then**
   |     |    **break**;
   |    **end**
**end**
**return** the predicted label $\hat{\mathbf{y}}$

---

**Update P, fix H.** Substitute the $\mathbf{H}$ obtained from the last iteration to Eq. (6), and let $\hat{\mathbf{M}} = \mathbf{OH}$, we can obtain the following optimization problem.

$$\min_{\mathbf{P}} \|\mathbf{MP} - \hat{\mathbf{M}}\|_F^2, \quad s.t. \ \mathbf{P1}_q = \mathbf{1}_q, \ \mathbf{P}^\top \mathbf{1}_q = \mathbf{1}_q, \ \mathbf{P} \in \{\mathbf{0}, \mathbf{1}\}^{q \times q}, \tag{12}$$

Due to the binary constraints on $\mathbf{P}$ , direct optimization is infeasible. We solve this assignment problem efficiently using the Hungarian algorithm via MATLAB's *matchpairs* function [44].

**Update W, fix R.** The subproblem regarding $\mathbf{W}$ can be obtained:

$$\min_{\mathbf{W}} \|\psi(\mathbf{X})^\top \mathbf{W} - \mathbf{R}\|_F^2 + \beta \|\mathbf{W}\|_{2,1} \ \Leftrightarrow \ \min_{\mathbf{W}} \|\psi(\mathbf{X})^\top \mathbf{W} - \mathbf{R}\|_F^2 + \beta \operatorname{tr}\left(\mathbf{W}^\top \mathbf{DW}\right), \tag{13}$$

where the diagonal elements of $\mathbf{D}$ are computed as $D_{ii} = 1/\sqrt{\|w_i\|_2^2}(\forall i = 1, 2, 3, ..., h)$. Taking the derivative of Eq. (13) w.r.t. $\mathbf{W}$ and setting it to 0, we can get the following equation:

$$\mathbf{W} = (\psi(\mathbf{X})\psi(\mathbf{X})^\top + \beta \mathbf{D})^{-1} \psi(\mathbf{X})\mathbf{R}, \tag{14}$$

**Update R, fix W.** We solve the optimization involving the non-convex $\ell_{2,1}$ norm and non-negative constraint $\mathbf{R} \geq 0$ by applying the Lagrange multiplier method, yielding the following Lagrangian function:

$$\min_{\mathbf{R}} \|\mathbf{C} \odot (\mathbf{Y} - \mathbf{R})\|_F^2 + \|\psi(\mathbf{X})^\top \mathbf{W} - \mathbf{R}\|_F^2 + \alpha \|\mathbf{Q}\|_1 + \mu/2 \|\mathbf{Y} - \mathbf{R} - \mathbf{Q}\|_F^2 - \operatorname{tr}\left(\boldsymbol{\Theta}\mathbf{R}^\top\right), \tag{15}$$

where $\mu$ is a number large enough and $\boldsymbol{\Theta}$ represents the Lagrange multiplier. Taking the derivative of Eq. (15) and setting the derivative to zero. Then based on condition of Karush-Kuhn-Tucker (KKT), it can be given that: $\boldsymbol{\Theta} \odot \mathbf{R} = \mathbf{0}$, that is, $\theta_{ij} r_{ij} = 0$. Fix $\mathbf{Q}$, then take the derivative of Eq. (15) w.r.t. $\mathbf{R}$ and setting it to 0, we can get the following equation: We can get the update rules for $\mathbf{R}$:

$$r_{ij}^{(t+1)} = r_{ij}^{(t)} \frac{\mathbf{B}_{ij}}{\mathbf{A}_{ij} + eps}, \tag{16}$$

where $\mathbf{A}_{ij} = (2\mathbf{R} + 2\mathbf{C} \odot \mathbf{R} \odot \mathbf{C} + \mu(\mathbf{R} + \mathbf{E}))_{ij}$ and $\mathbf{B}_{ij} = (2\psi(\mathbf{X})^\top \mathbf{WW} + 2\mathbf{C} \odot \mathbf{Y} \odot \mathbf{C} + \mu\mathbf{Y})_{ij}$, With $\mathbf{R}$ fixed, we can get the sub-optimization of $\mathbf{Q}$:

$$\min_{\mathbf{Q}} \mathcal{L}(\mathbf{Q}) = \alpha \|\mathbf{Q}\|_1 + \frac{\mu}{2} \|\mathbf{Y} - \mathbf{R} - \mathbf{Q}\|_F^2, \tag{17}$$

which is a typical LASSO regression problem [45], and we apply PGD algorithm to optimize it. The proximal operator of Eq. (17) is:

$$\text{prox}_{th(\cdot)}\left(\mathbf{Q}\right) = arg \min_{\mathbf{Q}} \|\mathbf{Q} - \mathbf{Z}\|_F^2 + \frac{\alpha}{\mu\mathbf{L}}\|\mathbf{Q}\|_1 , \tag{18}$$

where $\mathbf{Z} = \mathbf{Q}^t - \frac{1}{\mathbf{L}}\nabla\mathcal{L}(\mathbf{Q}^{(t)})$, $\mathbf{Q}^{(t)}$ represents the solution of $\mathbf{Q}$ from the $t$-th iteration, and $\nabla\mathcal{L}(\mathbf{Q})$ is the gradient of the objective function $\mathcal{L}(\mathbf{Q})$, $\mathbf{L}$ is the Lipschitz constant of $\nabla\mathcal{L}(\mathbf{Q})$ and $t$ denotes the number of iteration. Eq. (18) can be iteratively updated by the soft-thresholding operator [46]:

$$q_{ij}^{(t+1)} = Soft[q_{ij}^{(t)} - \frac{1}{\mathbf{L}}\nabla\mathcal{L}(q_{ij}^t, \frac{\alpha}{\mu\mathbf{L}})], \tag{19}$$

where $Soft[b, \nu] = sign(b)max\{|b| - \nu, 0\}$. In addition, the Lipschitz constant of $\nabla\mathcal{L}(\mathbf{Q})$ is 1, so we set $\mathbf{L} = 1$. The overall pseudo code of CAPML is summarized in Algorithm 1 .

# 3 Experiment

## 3.1 Experimental Setup

**Datasets** To evaluate the generalization performance of our proposed CAPML approach, we conducted experiments on 10 datasets, including 6 real-world PML datasets[49] and 18 synthetic PML datasets generated from seven multi-label datasets[47, 48]. For clarity, the detailed characteristics of these datasets are shown in Table 1. Specifically, the synthetic datasets are derived from multi-label datasets by adding noise to the labels. For each instance, a portion of irrelevant labels is randomly picked as candidate labels along with the relevant ones. Taking the $birds$ dataset as an example, which originally has 1.01 ground-truth labels per instance ($avg.\#GLs$), we created three noisy variants with 3, 4, and 5 candidate labels per instance ($avg.\#CLs$) by randomly injecting approximately 1.99, 2.99, and 3.99 false positive labels per instance, respectively.

Table 1: Characteristics of experimental data sets.

| Datsets | #Instances | #Dim | #Classes | avg.#CLs | avg.#GLs | Domain |
|---|---|---|---|---|---|---|
| Mirflickr | 10433 | 100 | 7 | 3.35 | 1.77 | Images[1] |
| Music_emotion | 6833 | 98 | 11 | 5.29 | 2.42 | Music[1] |
| Msic_style | 6839 | 98 | 10 | 6.04 | 1.44 | Music[1] |
| YeastBP | 6139 | 6139 | 217 | 5.93 | 5.54 | Biology[1] |
| YeastCC | 6139 | 6139 | 50 | 1.39 | 1.35 | Biology[1] |
| YeastMF | 6139 | 6139 | 39 | 1.04 | 1.01 | Biology[1] |
| emotions | 593 | 72 | 6 | 3, 4, 5 | 1.86 | Music[2] |
| birds | 645 | 260 | 19 | 3, 4, 5 | 1.01 | Audio[2] |
| medical | 978 | 1449 | 45 | 5, 7, 9 | 1.25 | Text[2] |
| image | 2000 | 294 | 5 | 2, 3, 4 | 1.23 | Images[2] |
| yeast | 2417 | 103 | 14 | 7, 9, 11 | 4.24 | Biology[2] |
| corel5k | 5000 | 499 | 374 | 7, 9, 11 | 3.52 | Images[2] |

[1] http://palm.seu.edu.cn/zhangml/, [2] http://mulan.sourceforge.net/datasets.html

**Comparison approaches** The performance of CAPML is compared with seven state-of-the-art methods, the following is a brief introduction for each comparison approach:

- fPML [16] [2019]: fPML removes noise by decomposing the candidate label matrix into two low-rank matrices and utilizing the resulting low-error approximation. [configuration: $\lambda_1 = 1, \lambda_2 = 1, \lambda_3 = 10$].

- PARTICLE(PAR-MAP and PAR-VLS) [12] [2020]: A two-stage PML approach that refines candidate labels through label propagation and builds distinct predictive models [suggested configuration: $k = 10, \alpha = 0.9, thr = 10.9$].

- PML-NI [5] [2021]: Considering that fuzzy features may produce noise labels, the prediction model matrix is decomposed into truth label prediction and noise label prediction [configuration: $\lambda = 10, \beta = \gamma = 0.5, max\_iter = 500$].

- PAMB [29] [2023]: PAMB uses ECOC techniques to convert PML into a binary classification problem, avoiding the error-prone estimation of individual label confidences [configuration: $z = avg.\#CLs, L = 100\log_2(q)$].

Table 2: Comparision of CAPML with other state-of-the-art PML approaches on *Average Precision* (mean±std), where the best experimental performance (the larger the better) is shown in boldface.

| Data Sets | avg.#CLs | CAPML | FBD-PML | LENFN | PAMB | PML-NI | PARTICLE | FPML |
|---|---|---|---|---|---|---|---|---|
| Mirflickr | 3.35 | **0.820±0.008** | 0.815±0.007 | 0.800±0.009 | 0.791±0.019 | 0.786±0.009 | 0.813±0.136 | 0.814±0.009 |
| Music emotion | 5.29 | **0.628±0.010** | 0.607±0.011 | 0.608±0.010 | 0.626±0.011 | 0.608±0.012 | 0.506±0.016 | 0.458±0.015 |
| Music style | 6.04 | 0.743±0.016 | 0.740±0.017 | **0.745±0.014** | 0.741±0.007 | 0.739±0.015 | 0.657±0.012 | 0.566±0.090 |
| YeastBP | 5.93 | **0.443±0.015** | 0.406±0.021 | 0.423±0.017 | 0.356±0.022 | 0.404±0.022 | 0.168±0.016 | 0.328±0.012 |
| YeastCC | 1.39 | **0.609±0.023** | 0.584±0.011 | 0.603±0.019 | 0.556±0.024 | 0.454±0.025 | 0.348±0.016 | 0.458±0.032 |
| YeastMF | 1.04 | **0.495±0.021** | 0.431±0.017 | 0.484±0.023 | 0.405±0.014 | 0.418±0.016 | 0.228±0.012 | 0.326±0.009 |
| | 3 | **0.807±0.039** | 0.783±0.008 | 0.783±0.035 | 0.804±0.017 | 0.777±0.028 | 0.747±0.035 | 0.663±0.020 |
| emotions | 4 | **0.787±0.027** | 0.765±0.006 | 0.761±0.030 | 0.783±0.036 | 0.749±0.034 | 0.739±0.033 | 0.651±0.016 |
| | 5 | **0.756±0.032** | 0.751±0.005 | 0.746±0.033 | 0.749±0.026 | 0.680±0.039 | 0.702±0.037 | 0.654±0.029 |
| | 3 | **0.627±0.058** | 0.625±0.006 | 0.621±0.063 | 0.589±0.052 | 0.617±0.057 | 0.379±0.046 | 0.381±0.037 |
| birds | 4 | **0.590±0.057** | 0.586±0.003 | 0.584±0.050 | 0.564±0.044 | 0.572±0.041 | 0.419±0.046 | 0.373±0.020 |
| | 5 | **0.589±0.048** | 0.573±0.026 | 0.568±0.027 | 0.495±0.029 | 0.564±0.034 | 0.372±0.047 | 0.371±0.018 |
| | 5 | 0.876±0.027 | 0.864±0.024 | **0.878±0.022** | 0.815±0.012 | 0.866±0.024 | 0.754±0.047 | 0.838±0.025 |
| medical | 7 | **0.866±0.031** | 0.856±0.031 | 0.864±0.018 | 0.796±0.031 | 0.835±0.036 | 0.741±0.049 | 0.832±0.029 |
| | 9 | **0.852±0.033** | 0.842±0.020 | 0.851±0.028 | 0.771±0.011 | 0.798±0.031 | 0.715±0.022 | 0.817±0.019 |
| | 2 | **0.814±0.021** | 0.778±0.026 | 0.777±0.023 | 0.798±0.024 | 0.770±0.020 | 0.743±0.070 | 0.711±0.018 |
| image | 3 | **0.792±0.018** | 0.745±0.031 | 0.745±0.025 | 0.748±0.019 | 0.732±0.024 | 0.725±0.084 | 0.696±0.023 |
| | 4 | **0.759±0.022** | 0.691±0.011 | 0.671±0.027 | 0.711±0.026 | 0.653±0.011 | 0.668±0.091 | 0.670±0.022 |
| | 7 | 0.760±0.018 | 0.734±0.007 | 0.756±0.020 | **0.761±0.014** | 0.746±0.017 | 0.754±0.013 | 0.732±0.016 |
| yeast | 9 | **0.755±0.021** | 0.725±0.022 | 0.738±0.019 | 0.750±0.013 | 0.725±0.016 | 0.744±0.011 | 0.730±0.013 |
| | 11 | **0.748±0.013** | 0.704±0.015 | 0.719±0.018 | 0.741±0.013 | 0.692±0.013 | 0.728±0.013 | 0.698±0.011 |
| | 7 | **0.306±0.015** | 0.274±0.015 | 0.282±0.016 | 0.239±0.017 | 0.279±0.013 | 0.254±0.003 | 0.266±0.004 |
| corel5k | 9 | **0.303±0.017** | 0.267±0.015 | 0.273±0.015 | 0.230±0.008 | 0.273±0.013 | 0.234±0.004 | 0.264±0.001 |
| | 11 | **0.299±0.017** | 0.266±0.018 | 0.265±0.015 | 0.228±0.019 | 0.266±0.015 | 0.230±0.014 | 0.262±0.005 |

Table 3: Comparision of CAPML with other state-of-the-art PML approaches on *Ranking Loss* (mean±std), where the best experimental performance (the smaller the better) is shown in boldface.

| Data Sets | avg.#CLs | CAPML | FBD-PML | LENFN | PAMB | PML-NI | PARTICLE | FPML |
|---|---|---|---|---|---|---|---|---|
| Mirflickr | 3.35 | **0.110±0.005** | 0.126±0.006 | 0.121±0.004 | 0.112±0.038 | 0.126±0.007 | 0.127±0.103 | 0.115±0.006 |
| Music emotion | 5.29 | 0.236±0.007 | 0.249±0.012 | 0.245±0.012 | **0.234±0.007** | 0.246±0.008 | 0.362±0.014 | 0.410±0.005 |
| Music style | 6.04 | **.0.135±0.010** | 0.139±0.024 | 0.140±0.012 | 0.136±0.005 | 0.137±0.010 | 0.221±0.010 | 0.317±0.033 |
| YeastBP | 5.93 | **0.203±0.009** | 0.271±0.013 | 0.253±0.009 | 0.230±0.011 | 0.220±0.011 | 0.404±0.033 | 0.415±0.057 |
| YeastCC | 1.39 | **0.167±0.015** | 0.194±0.017 | 0.191±0.012 | 0.221±0.021 | 0.210±0.022 | 0.480±0.014 | 0.342±0.019 |
| YeastMF | 1.04 | **0.218±0.017** | 0.262±0.011 | 0.232±0.013 | 0.244±0.013 | 0.226±0.018 | 0.533±0.010 | 0.373±0.019 |
| | 3 | 0.162±0.036 | 0.181±0.018 | 0.182±0.033 | **0.160±0.022** | 0.188±0.029 | 0.250±0.035 | 0.474±0.027 |
| emotions | 4 | **0.170±0.029** | 0.197±0.013 | 0.194±0.029 | 0.178±0.031 | 0.211±0.027 | 0.263±0.029 | 0.446±0.027 |
| | 5 | **0.205±0.032** | 0.213±0.006 | 0.251±0.032 | 0.210±0.015 | 0.276±0.039 | 0.306±0.034 | 0.452±0.037 |
| | 3 | **0.172±0.028** | 0.176±0.035 | 0.183±0.007 | 0.196±0.042 | 0.177±0.033 | 0.322±0.033 | 0.333±0.041 |
| birds | 4 | 0.195±0.042 | **0.195±0.034** | 0.211±0.026 | 0.204±0.028 | 0.205±0.034 | 0.326±0.027 | 0.341±0.020 |
| | 5 | **0.200±0.036** | 0.207±0.038 | 0.223±0.012 | 0.229±0.025 | 0.219±0.036 | 0.359±0.036 | 0.328±0.018 |
| | 5 | **0.029±0.011** | 0.049±0.018 | 0.038±0.015 | 0.050±0.002 | 0.040±0.012 | 0.090±0.019 | 0.052±0.007 |
| medical | 7 | **0.032±0.010** | 0.047±0.016 | 0.045±0.016 | 0.062±0.019 | 0.052±0.013 | 0.111±0.022 | 0.058±0.009 |
| | 9 | **0.038±0.013** | 0.051±0.016 | 0.054±0.016 | 0.099±0.023 | 0.061±0.015 | 0.122±0.016 | 0.057±0.010 |
| | 2 | **0.155±0.021** | 0.184±0.013 | 0.190±0.025 | 0.177±0.022 | 0.194±0.019 | 0.230±0.060 | 0.239±0.019 |
| image | 3 | **0.183±0.020** | 0.206±0.011 | 0.214±0.028 | 0.217±0.015 | 0.230±0.024 | 0.261±0.070 | 0.254±0.018 |
| | 4 | **0.213±0.022** | 0.280±0.017 | 0.287±0.019 | 0.255±0.025 | 0.303±0.010 | 0.328±0.095 | 0.280±0.022 |
| | 7 | **0.173±0.013** | 0.187±0.005 | 0.180±0.014 | 0.214±0.008 | 0.184±0.012 | 0.182±0.010 | 0.186±0.014 |
| yeast | 9 | **0.177±0.015** | 0.198±0.019 | 0.192±0.018 | 0.211±0.007 | 0.202±0.016 | 0.189±0.009 | 0.191±0.012 |
| | 11 | **0.183±0.012** | 0.225±0.015 | 0.219±0.010 | 0.231±0.014 | 0.199±0.010 | 0.195±0.010 | 0.216±0.008 |
| | 7 | 0.176±0.007 | **0.175±0.007** | 0.223±0.012 | 0.310±0.034 | 0.215±0.008 | 0.317±0.008 | 0.255±0.013 |
| corel5k | 9 | **0.184±0.007** | 0.192±0.016 | 0.225±0.010 | 0.315±0.049 | 0.227±0.009 | 0.324±0.008 | 0.262±0.013 |
| | 11 | **0.189±0.006** | 0.203±0.012 | 0.231±0.009 | 0.319±0.059 | 0.232±0.008 | 0.330±0.007 | 0.268±0.011 |

- PML-LENFN [50] [2024]: PML-LENFN improves label quality by jointly analyzing local (neighbor) and global (distant) sample relationships, paired with a hybrid classifier combining linear and nonlinear components [ $\lambda_1 = 10^{-5}$, $\lambda_2 = 1$, $\lambda_3 = 10^{-5}$, $\lambda_4 = 10^{-5}$].

- FBD-PML [19] [2025]: FBD-PML performs manifold alignment by simultaneously learning the prototypes of features and labels to achieve the smooth assumption. [configuration: $\lambda_1 = 10^{-4}$, $\lambda_2 = 1$, $\lambda_3 = 10^{-3}$, $\lambda_4 = 10^{-3}$, $\lambda_5 = 10^{-2}$]

For our CAPML approach $\lambda$ is set to 0.5, $\alpha$ and $\beta$ are searched in {0.001, 0.01, 0.1, 1, 10, 100}. For the feature mapping function $\psi(\cdot)$, we employ the Gaussian kernel with bandwidth parameter set to the average pairwise distance between samples.

## 3.2 Experimental Metrics and Results

In our experiment, we evaluate the performance of CAPML and other state-of-the-art baselines using five multi-label metrics: *Hamming loss*, *Ranking loss*, *One-error*, *Coverage*, and *Average precision*.

Details of these metrics can be found in 47. And we apply ten-fold cross-validation on each PML dataset and report the mean and standard deviation for all eight comparison approaches. Due to page limitations, our experimental evaluation focuses primarily on two representative metrics: *Average Precision* and *Ranking Loss*, which are complementary metrics providing comprehensive insights into both ranking quality and classification accuracy. Results for the other three metrics are provided in the Appendix. Complete results are presented in Tables 2 and 3, respectively.

Moreover, to verify the statistical significance of CAPML's performance advantages, we summarize *win/tie/loss* counts across all evaluation metrics against each competing approach at the 0.05 significance level, which is shown in Table 4. From the experimental results and subsequent statistical analysis, we can draw several significant conclusions:

- Across all evaluation metrics, our method achieves state-of-the-art performance in 86% of cases over the entire collection of 24 datasets. From Table 2 and 3, it can be observed that in 87.5% and 83.3% of the datasets, the approach consistently outperforms in the *Average Precision* and *Ranking Loss* metrics. The statistical advantages persist across diverse data characteristics—from high-dimensional biological datasets ($YeastBP, YeastCC$) to low-dimensional multimedia collections (Music, Image)—suggesting the method's superior generalization performance. Even on the challenging corel5k datasets, where label spaces are particularly sparse, CA-PML maintains its statistical edge over all competitors.

- From Table 4, CAPML demonstrates convincing statistical dominance, winning in 607 out of 720 total comparisons (84.3%) while achieving statistical ties in all remaining cases. The smallest, though still significant, improvements are observed against LENFN, which indicates our effective guidance of label reliability indicator.

- FBD-PML represents a notable baseline as it is also a prototype-based approach. Despite this similarity, CAPML consistently outperforms FBD-PML, underscoring the efficacy of mutually prototype alignment strategy and stage-wise gradual label disambiguation.

Table 4: Win/tie/loss counts of pairwise t-test (at 0.05 aignificance level) on CAPML against others

| CAPML **against** | FBD-PML | LENFN | PAMB | PML-NI | PARTICLE | FPML |
|---|---|---|---|---|---|---|
| *Hamming loss* | 21/3/0 | 16/8/0 | 21/3/0 | 20/4/0 | 20/4/0 | 18/6/0 |
| *Ranking loss* | 17/7/0 | 20/4/0 | 18/6/0 | 21/3/0 | 23/1/0 | 23/1/0 |
| *One-error* | 22/2/0 | 23/1/0 | 18/6/0 | 23/1/0 | 20/4/0 | 23/1/0 |
| *Coverage* | 20/4/0 | 17/7/0 | 18/6/0 | 19/5/0 | 23/1/0 | 22/2/0 |
| *Average precision* | 19/5/0 | 16/8/0 | 16/8/0 | 22/2/0 | 22/2/0 | 23/1/0 |
| **In Total** | **99/21/0** | **93/27/0** | **91/29/0** | **105/15/0** | **108/12/0** | **109/11/0** |

## 4  Further Analysis

**Parameter Sensitivity**  We assess CAPML's robustness through parameter sensitivity analysis, examining how hyperparameters $\alpha$ and $\beta$ influence the model's Average Precision performance. We vary each parameter individually while keeping the other fixed, with the results visualized as paired bar charts in Figure 2 for direct comparison. Figure 2 shows remarkable robustness of CAPML to parameter variations, with Average Precision remaining stable across a wide range of values. This stability is particularly evident for parameter $\alpha$, where performance fluctuations are minimal within the reasonable range of [0.01, 100]. Similarly, when varying $\beta$, the model maintains consistent performance with only slight degradation at extreme values.

**Computational Complexity**  The algorithm complexity is $O(T_0 dnq + T_1(dq^2 + q^3) + T_2(nhq + h^2q + d^3 + nq))$. Stage one in Algorithm 1 involves SVD of the $d \times q$ matrix $O^T(MP)$ with complexity $O(dq^2)$ for the orthogonal Procrustes problem in Eq. (11), plus $O(q^3)$ for the Hungarian algorithm solving the assignment problem in Eq. (12), totaling $O(dq^2 + q^3)$ per iteration. Stage two involves $O(nhq + h^2q)$ for matrix operations in W-update including the inversion of $h \times h$ matrix, and $O(nq)$ for both R and Q updates involving element-wise operations. The dominant computational cost depends on the relative magnitudes of $d$, $h$, $n$, and $q$, but typically the $O(nhq)$ term dominates when datasets are large, making the method practically scalable.

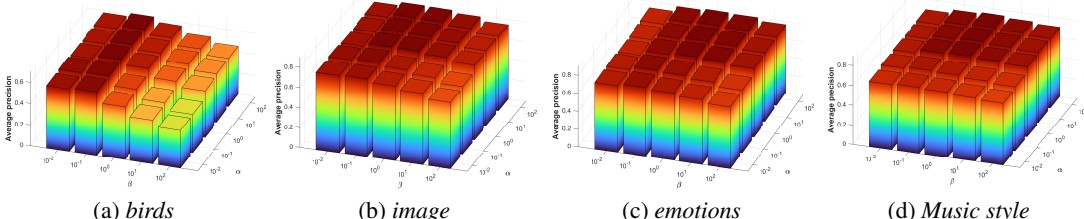

(a) *birds*       (b) *image*       (c) *emotions*       (d) *Music style*

Figure 2: AP variations with parameters $\alpha$ and $\beta$ on birds(avg.#CLs=2), emotions(avg.#CLs=3), image(avg.#CLs=2) and music style datasets.

**Ablation Study** To investigate the contribution of each key component in CAPML, we conduct ablation studies by comparing our full model with two variants: (1) **CAPML-ED**, which replaces our entropy-regularized fuzzy clustering with direct Euclidean distance between instances and prototypes computed from candidate label set to derive membership degrees; (2) **CAPML-NR**, which removes the orthogonal rotation matrix **H** from the prototype alignment process. (3) **CAPML-NA**, which sets permutation matrix P to identity matrix, removing prototype alignment. (4)**CAPML-CW**, which sets label enhancement indicator matrix C to all-ones, removing confidence indicator. Figure 3 presents the comparative results across the seven of all benchmark datasets on *Average Precision* and *Ranking loss*. These results validate the effectiveness of combining entropy-regularized clustering with orthogonal transformation for prototype learning.

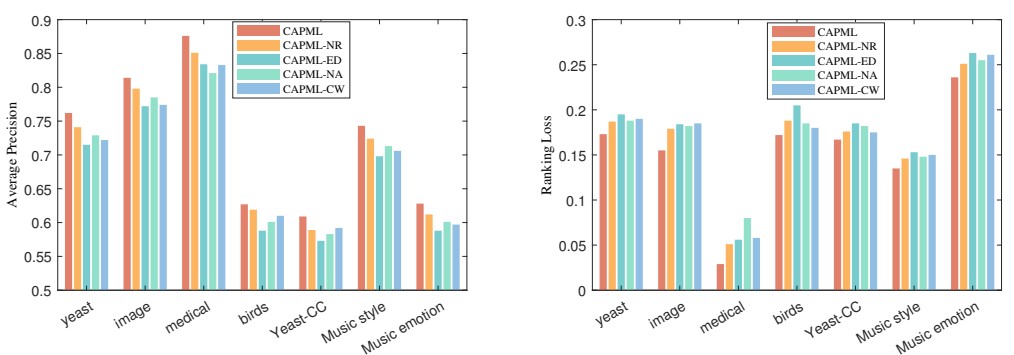

Figure 3: AP variations with parameters $\alpha$ and $\beta$ on yeast(avg.#CLs=7), image(avg.#CLs=2), medical(avg.#CLs=5), birds(avg.#CLs=2), YeastCC, music style and music emotion datasets.

## 5 Conclusions

This paper introduces CAPML, a novel PML approach addressing label disambiguation through mutual prototype alignment. Unlike methods relying on noisy candidate labels alone, we align unsupervised prototypes capturing clean data structure with supervised prototypes containing semantic information. Through permutation matrices and orthogonal rotation, we transform fuzzy memberships into reliable confidence indicators operating external to classifier learning. This dual-prototype framework, combined with confidence-aware disambiguation and sparse regularization, effectively identifies true labels under challenging noise conditions. Comprehensive experiments demonstrate CAPML's significant advantages over state-of-the-art methods.

## 6 limitations

Despite these promising results, CAPML assumes fuzzy clustering discovers structure aligned with label semantics. Performance may degrade when high intra-class variability fragments categories into multiple clusters, when similar features cause distinct labels to merge, or when label counts considerably exceed natural cluster structures.

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
