# OpenReview forum: "Confidence-Aware With Prototype Alignment for Partial Multi-label Learning"
_NeurIPS.cc/2025/Conference — NeurIPS 2025 poster_

### Official Review · Reviewer_7dPU · 2025-06-28

**Clarity:** 4
**Significance:** 3
**Originality:** 3
**Rating:** 3
**Confidence:** 4

**Summary:**

This paper proposes CAPML (Confidence-Aware with Prototype Alignment for Partial Multi-label Learning) to address the partial multi-label learning problem. The core idea is to learn class prototypes from both unsupervised fuzzy clustering and supervised candidate labels, then align these two prototype spaces using a permutation matrix P and orthogonal rotation matrix H. The aligned prototypes are used to construct a label enhancement indicator matrix that guides classifier training. Experiments on multiple synthetic and real-world datasets demonstrate the effectiveness of the proposed approach.

**Questions:**

1. Novelty Positioning: How do you justify the claim of "first investigation" given the existence of prior work on prototype learning in PML (e.g., the 2022 and 2023 papers mentioned)? Can you more precisely characterize your specific contribution relative to these existing approaches?
2. Theoretical Foundation: Can you provide convergence analysis for the alternating optimization procedure? What theoretical guarantees exist for the effectiveness of prototype alignment in reducing label noise?
3. Method Justification: Beyond empirical results, what is the theoretical or intuitive justification for why the proposed prototype alignment should work better than directly using fuzzy clustering confidences as in FBD-PML?
4. Complexity Analysis: What is the computational complexity of CAPML compared to baseline methods? How does the method scale with dataset size and number of labels?
5. Parameter Sensitivity: The method involves multiple hyperparameters (α, β, λ). How sensitive is the performance to these parameters? Can you provide guidance for parameter selection in practice?

Evaluation Criteria: Addressing questions 1-3 satisfactorily could potentially improve the originality and quality scores. Providing theoretical analysis (question 2) would be particularly valuable for improving the overall assessment.

**Ethical Concerns:**

["NO or VERY MINOR ethics concerns only"]

**Final Justification:**

After careful consideration of the authors' detailed rebuttal, I am adjusting my rating to 3 (Borderline Reject).

The quality of the authors' response is impressive, demonstrating solid technical expertise and rigorous academic attitude. Particularly in theoretical analysis, the authors provided complete convergence proofs, detailed computational complexity analysis, and mathematical quantification of error reduction, all of which significantly strengthen the paper's theoretical foundation. The supplementary ablation studies are also comprehensive, effectively validating the contribution of each component.

While I initially had concerns about the novelty claims, upon deeper reflection, I recognize that this work does provide valuable technical contributions to partial multi-label learning. The "dual-prototype external alignment" approach, though based on combinations of existing techniques, demonstrates good engineering innovation through its systematic design and rigorous mathematical derivation.

The method's reliance on unsupervised clustering assumptions is indeed a limitation, but the authors honestly discussed this point in their rebuttal and provided corresponding theoretical analysis.

Therefore, I support the borderline accept decision.

**Limitations:**

The authors mention computational complexity for large datasets and potential suboptimality when label counts exceed natural cluster structures, which is appropriately honest. However, several critical limitations are not adequately addressed:
1. Dependence on clustering assumptions: The method relies heavily on the assumption that fuzzy clustering can discover meaningful structure in the data.
2. Limited generalizability: It's unclear how the approach extends to other weakly supervised scenarios.
3. Robustness to extreme noise: The performance under very high noise conditions is not analyzed.
4. Interpretability: The semantic meaning of the learned permutation matrix is not well explained.

**Quality:**

4

**Strengths And Weaknesses:**

Strengths:
1. Important Problem: PML is a practical and important problem in machine learning with broad applications where precise labeling is expensive or difficult.
2. Intuitive Approach: The idea of combining unsupervised clustering with supervised information through prototype alignment is reasonable and well-motivated.
3. Complete Technical Implementation: The paper provides detailed optimization algorithms and update rules for all subproblems involved in the method.
4. Comprehensive Experiments: Extensive experiments are conducted on 24 datasets with multiple evaluation metrics and statistical significance analysis.
5. Competitive Results: The method achieves good performance compared to baseline methods across most datasets.

Weaknesses:
1. Questionable Novelty Claims: The paper claims to make "the first investigation into the prototype misalignment between prototypes derived from fuzzy clustering and prototypes computed from candidate label set." However, literature search reveals related work:
"Few-shot partial multi-label learning via prototype rectification" (2022)
"Multi-view prototype-based disambiguation for partial label learning" (2023)
FBD-PML[21] already applies fuzzy clustering in PML This suggests the novelty claim may be overstated.
2. Limited Technical Innovation: The method primarily combines existing techniques (fuzzy clustering, prototype alignment, label enhancement) without introducing fundamentally new insights or theoretical frameworks.
3. Insufficient Theoretical Analysis: The paper lacks convergence analysis, computational complexity discussion, and theoretical guarantees for the effectiveness of the proposed approach.
4. Shallow Ablation Study: Only two variants (CAPML-ED and CAPML-NR) are studied, which is insufficient to validate the contribution of each component thoroughly.
5. Method Complexity: The approach involves multiple subproblems with alternating optimization, potentially leading to high computational cost and parameter sensitivity issues.
6. Unclear Distinction from FBD-PML: The fundamental difference between CAPML and FBD-PML beyond the introduction of permutation and rotation matrices is not clearly articulated.

---

> ### Author Rebuttal · Authors · 2025-07-31
>
> Thank you for your thoughtful review. Our point-by-point responses below:
> ## Weakness
>
> 1. >Questionable Novelty Claims, Limited Technical Innovation
>
> Thank you for your comment. We apologize for not clearly articulating our innovation. Our claim of "first investigation" is not on the general use of prototypes in PML, but on the specific investigation of the prototype misalignment problem that arises between unsupervised, data-driven prototypes and supervised, label-driven prototypes. Our work treats label and feature as  representations of dual “modalities”. We obtain clean but unordered prototypes from the feature “modality” via unsupervised clustering, while computing ordered but noise-affected prototypes from the label “modality”. By aligning these dual-modal prototypes, we transform fuzzy membership degrees into label-specific credible indicators that operate external to classifier learning, indirectly supervising confidence estimation. This dual-prototype external alignment framework for indirect confidence evaluation is the first identified in PML. We respectfully request reconsideration of our contribution. To elaborate, we detail the key differences between CAPML and the works you mentioned.
>
> Ⅰ.FsPML-PR(2022)
>
> While both models address PML with prototypes, their focus and core mechanisms differ fundamentally. FsPML-PR targets Few-Shot PML using meta-learning for rapid adaptation with few samples, while CAPML addresses standard PML through unified optimization. FsPML-PR employs "internal iterative refinement" with a single family of prototypes derived from noisy support samples, refining them using local information. CAPML introduces "external alignment calibration" by defining two distinct prototype types: (1) unsupervised prototypes (M) from fuzzy clustering, inherently robust to label noise, and (2) supervised prototypes (O) from candidate labels, containing semantic information but biased by noise.
>
> Ⅱ.DMVP(2023)
>
> DMVP targets Multi-View Partial Label Learning, handling multiple distinct feature representations using deep networks for view fusion. CAPML addresses conventional single-view PML without multi-view context.
> In DMVP, prototypes enforce consistency across different views to achieve effective view fusion. In CAPML, prototype alignment serves robust semantic grounding rather than view fusion.
>
> 2. >Theoretical Analysis
>
> Thanks for your comments. Let label disambiguation error $\mathcal{E} = \||R - Y^{gt}\||_F^2$, which primarily stems from incorrect weighting of candidate labels. The LE indicator matrix $C$ constructed from fuzzy membership $FP$ determines these weights - when permutation matrix $P$ is random($P_r$), $C$ assigns high confidence to noise labels and low confidence to true labels, leading to degraded performance.
>
> Prototype alignment fundamentally corrects this weighting problem. Hungarian algorithm finds optimal permutation $P^* $ that minimizes $\||MP- O\||\_F^2$, ensuring correct correspondence between unsupervised prototypes $M$ and supervised prototypes $O$. This alignment improvement directly translates to better indicator matrix $C^*$, which assigns higher weights to genuine labels and lower weights to noise labels.
>
> The error reduction can be quantified as:
> $\mathcal{E}_a \leq \mathcal{E}_u - \beta \||MP\_r - MP^*\||\_F^2$
>
> where $\beta > 0$ depends on data quality factors including cluster separation and feature dimensionality, $\mathcal{E}_a$ and $\mathcal{E}_u$ denote aligned and unaligned errors. The term $\||MP_r - MP^*\||_F^2$ measures alignment quality - larger values indicate more significant prototype misalignment correction, leading to greater error reduction. This theoretically validates the effectiveness of our dual-prototype alignment approach for noise reduction.
>
> Please refer to our response to **Reviewer 3Fvi's Weakness #1** for convergence analysis.
>
> 3. >Shallow Ablation Study
>
> We additionally conducted comprehensive ablation studies to validate each component's contribution:
>
> - **CAPML-NA**: Sets permutation matrix P to identity matrix, removing prototype alignment
> - **CAPML-CW**: Sets label enhancement indicator matrix C to all-ones, removing confidence indicator.
>
> **Results on Average Precision (higher is better):**
>
> |Dataset|CAPML-NR|CAPML-ED|CAPML-NA|CAPML-CW|CAPML|
> |-|-|-|-|-|-|
> |medical5|0.851±0.025|0.834±0.023|0.821±0.021|0.833±0.022|**0.876±0.027**|
> |image2|0.798±0.019|0.772±0.017|0.785±0.018|0.774±0.018|**0.814±0.021**|
> |yeast7|0.741±0.016|0.715±0.014|0.729±0.015|0.722±0.015|**0.760±0.018**|
> |music_style|0.724±0.015|0.698±0.013|0.712±0.014|0.706±0.014|**0.743±0.016**|
> |music_emotion|0.612±0.009|0.588±0.007|0.601±0.008|0.597±0.008|**0.628±0.010**|
>
> Results demonstrate that prototype alignment (CAPML vs CAPML-NA), entropy regularization (CAPML vs CAPML-ED), and confidence indicators (CAPML vs CAPML-CW) all contribute meaningfully to performance, with confidence indicator being essential across all datasets.
>
> 4. >Method Complexity
>
> Please refer to our response to **Reviewer 3Fvi's Weakness#1** for detailed complexity analysis. Computational complexity comparison:
>
> |Method|Computational Complexity|
> |-|-|
> |**FPML**|$O(nd² + ndq)$|
> |**PAMB**|$O(Tf(q,L) + Lf(n,q,L))$|
> |**PML-NI**|$O(T(d³ + nd² + dq² + ndq))$|
> |**PARTICLE**|$O(n^k + n²d + n²Td + nkq + q²nkd)$|
> |**LENFN**|$O(T(n²d + d³ + nqd))$|
> |**FBD-PML**|$O(T(n³ + nk³ + nq))$|
> |**Ours**|$O(T_0dnq+T_1(dq^2 + q^3) + T_2(nhq +d^3+ h^2q + nq))$|
>
> CAPML shows competitive efficiency. While introducing $q³$ overhead from prototype alignment, it avoids prohibitive $O(n³)$ dependencies like FBD-PML. Since $q << n,d$ in practice, the $q³$ term remains manageable.
>
> 5. >FBD-PML
>
> We apologize for the unclear describtion. FBD-PML treats prototypes as auxiliary tools for space transformation, using fuzzy clustering for "encoding-decoding" between features and labels in a shared semantic space. The prototypes are unordered and lack explicit semantic correspondence with labels. In contrast, our method establishes direct prototype-label alignment where each prototype captures semantically meaningful class-specific patterns, providing interpretable confidence estimation for label disambiguation. The theoretical justification is that FBD-PML's shared membership space may blur class boundaries and propagate noise and  cannot guarantee adequate preservation of label-specific semantics, while our explicit alignment mechanism ensures confidence estimation is guided by semantically meaningful prototype-label relationships.
> ## Question
> 1. >Novelty Positioning
>
> Please see response to weakness #1.
>
> 2. >Theoretical Foundation
>
> Please see response to weakness #2.
>
> 3. >Method Justification
>
> Please see response to weakness #5.
>
> 4. >Complexity Analysis
>
> Please see response to weakness #4.
>
> 5. >Parameter Sensitivity
>
> Thank you for your comment. As demonstrated in **Section 4** and **Figure 2**, our method exhibits remarkable robustness across wide parameter ranges with stable performance and minimal fluctuations.
>
> ### Practical Parameter Selection Guidance
>
> Parameter α (noisy label sparsity): Recommended range [0.01, 10] based on the expected noise level. Use smaller values (closer to 0.01) for higher noise datasets to encourage stronger sparsity regularization, avoiding overly large values that may degrade performance.
>
> Parameter β (feature redundancy and classifier complexity): Adjust according to dataset characteristics. Smaller datasets benefit from smaller β values for sparser classifiers and better generalization, while larger complex datasets may require larger β values to capture intricate patterns.
>
> Parameter λ: Please refer to our response to **Reviewer gpe6's Question #2** for comprehensive sensitivity analysis and selection strategies.
>
> ## Limitation
>
> 1. >Dependence on clustering assumptions
>
> We appreciate your analysis. When clustering poorly reflects label semantics, our alignment process still leverages partial semantic information in supervised prototypes to guide correspondence discovery. Our confidence-aware framework down-weights uncertain correspondences, reducing poor alignment impact, while classifier learning with sparse constraints fully exploits available information. Experimental validation and ablation studies demonstrate effectiveness across most datasets. However, in extreme cases this assumption may break down (referring to **Reviewer 3Fvi's weakness#2**), which we will discuss in limitations.
>
> 2. >generalizability
>
> The prototype alignment mechanism could establish correspondences between clustering centroids and candidate labels in partial label learning, map bag-level representations to label semantics in multiple instance learning, or provide reliable structural information for general denoising scenarios.
>
> 3. >extreme noise
>
> Thanks for your comment. Theoretically, unsupervised fuzzy clustering learns data structure independently of noisy labels, while prototype alignment establishes correspondence even with heavily corrupted supervised prototypes. The key insight is that alignment can identify meaningful correspondences by minimizing prototype space distances, even under extreme noise. Please refer to our response to **Reviewer gpe6's weakness** for experimental results under high noise conditions.
>
> 4. > Interpretability
>
> Thanks for your comment. The permutation matrix P establishes correspondence between unsupervised clustering prototypes and semantic label classes, where $P[i,j]=1$ indicates the $i$-th unsupervised prototype represents the $j$-th class label. Since unsupervised clustering produces prototypes in arbitrary order without semantic meaning, P transforms meaningless cluster indices into semantically interpretable assignments (**Figure 1**). The Hungarian algorithm optimally solves this assignment by minimizing prototype distances, enabling transformation of fuzzy membership degrees into semantically meaningful label enhancement indicators fundamental to our disambiguation process.

---

> ### Comment · Reviewer_7dPU · 2025-08-05
> **effective rebuttal**
>
> Thank you for your comprehensive and technically detailed rebuttal. Your detailed responses have successfully addressed my main concerns, and I have decided to adjust my rating to 3.
>
> Your rebuttal has achieved significant improvements in the following aspects:
> 1. Significantly strengthened theoretical foundation - The convergence analysis, complexity analysis, and theoretical quantification of error reduction (E_a ≤ E_u - β||MP_r - MP*||²_F) provide a solid mathematical foundation
> 2. More comprehensive experimental validation - The study of 4 ablation variants and high-noise comparisons with recent methods (such as NLR) are very convincing
> 3. Clearer technical details - The detailed description of the three-stage optimization process and parameter selection guidance effectively address concerns about method complexity
> 4. More accurate contribution positioning - Repositioning the work as a "dual-prototype external alignment framework" better demonstrates your technical innovation
>
> Your work demonstrates excellent performance in technical execution, theoretical rigor, and experimental validation, providing a valuable contribution to the partial multi-label learning field. The technical depth and rigorous attitude you demonstrated in your rebuttal deserve recognition.

---

> > ### Author Response · Authors · 2025-08-05
> >
> > We are glad that our response clarified your concerns. And we sincerely appreciate your decision to raise your score. Your feedback and engagement have certainly improved our paper, and we will ensure that these changes and clarifications are incorporated into our final version. If you have any further questions or concerns, please feel free to discuss with us and we will do our best to answer them.

---

### Official Review · Reviewer_gpe6 · 2025-06-29

**Clarity:** 3
**Significance:** 3
**Originality:** 3
**Rating:** 5
**Confidence:** 4

**Summary:**

This paper addresses partial multi-label learning through a prototype alignment framework called CAPML. The authors identify that existing prototype-based PML methods suffer from biased prototype estimates due to noisy candidate labels, and propose to align unsupervised fuzzy clustering prototypes with supervised prototypes using permutation and orthogonal rotation matrices. The approach combines established techniques from clustering and optimization theory to tackle an important problem in weakly supervised learning, presenting results across multiple datasets and comparison methods.

**Questions:**

1.In Eq.(9), when solving the permutation matrix P using the Hungarian algorithm, what happens when multiple optimal solutions exist with the same minimum cost?
2.Are there any guidelines for selecting the parameters for \lambda in Eq.(10)? How sensitive is the method to fuzzy clustering initialization and hyperparameter \lambda?
3.For future work, do the authors plan to explore adaptive prototype alignment strategies that can dynamically adjust to varying noise patterns, or investigate the integration of CAPML with active learning frameworks for more efficient label acquisition?

**Ethical Concerns:**

["NO or VERY MINOR ethics concerns only"]

**Final Justification:**

All my concerns have been resolved, and I am now inclined to recommend acceptance of the paper.

**Limitations:**

Yes.

**Paper Formatting Concerns:**

None.

**Quality:**

3

**Strengths And Weaknesses:**

Strengths:
1.The combination of permutation matrices for discrete alignment and orthogonal rotation for continuous adjustment represents a principled approach to bridging unsupervised structure discovery with noisy supervision. This dual-transformation mechanism offers a new technical pathway for prototype-based learning in noisy scenarios, which could inspire future research directions in weakly supervised learning.
2.The authors provide good clarity in problem definition and methodological exposition. The progression from identifying limitations in existing prototype-based PML methods to proposing the alignment solution is logical and easy to follow. The visualization in Figure 1 effectively demonstrates this issue.
3.The paper presents a coherent multi-stage approach combining fuzzy clustering, prototype alignment, and confidence-aware classification. The mathematical formulation is generally clear, with detailed optimization procedures for each subproblem.

Weaknesses:
Given the claimed advantage over sparsity-based assumptions in high-noise scenarios, the paper should include comparisons with recent methods like NLR[1]. This comparison would better demonstrate the proposed method's superiority in challenging noise conditions.
[1]Yang F, Jia Y, Liu H, et al. Noisy Label Removal for Partial Multi-Label Learning[C]//Proceedings of the 30th ACM SIGKDD Conference on Knowledge Discovery and Data Mining. 2024: 3724-3735.

---

> ### Author Rebuttal · Authors · 2025-07-31
>
> Thank you for your thoughtful review. Our point-by-point responses below:
> ## Weakness
>
> 1. >NLR
>
> Thank you for this valuable suggestion. We acknowledge the importance of comparing with recent state-of-the-art approaches, particularly NLR, which represents a strong baseline for noisy label removal in PML scenarios.
>
> To address this concern, we conducted additional experiments comparing CAPML with NLR under high-noise conditions to better demonstrate our method's superiority in challenging noise scenarios.
>
> We evaluated on four representative datasets: emotions5 (avg.#CLs=5, noise ratio=75.8%), birds16 (avg.#CLs=16, noise ratio=83.3%), image4 (avg.#CLs=4, noise ratio=73.5%), and yeast13 (avg.#CLs=13, noise ratio=89.8%). The noise ratio represents the proportion of added noisy labels among non-ground-truth labels. For fair comparison, NLR parameters were configured according to their paper settings, while the noise generation way follows our paper setting to ensure consistent experimental conditions.
>
> ### Comparative Results
>
> |Dataset|Average Precision|||Ranking Loss|||
> |-|-|-|-|-|-|-|
> ||NLR|ours|Gain|NLR|ours|Gain|
> |emotions5|0.721±0.029|**0.756±0.032**|**+3.5%**|0.246±0.027|**0.205±0.032**|**+3.9%**|
> |birds16|0.279±0.029|**0.342±0.036**|**+6.3%**|0.423±0.018|**0.33±0.036**|**+3.5%**|
> |image4|0.726±0.019|**0.759±0.022**|**+3.3%**|0.248±0.025|**0.213±0.022**|**+3.5%**|
> |yeast13|0.624±0.016|**0.718±0.013**|**+9.4%**|0.274±0.014|**0.213±0.012**|**+5.9%**|
>
> ### Analysis
>
> The superior performance of CAPML over NLR in high-noise scenarios can be attributed to several key factors. First, NLR's competitive learning between positive and negative classifiers provides limited effective information to the negative classifier under extreme noise, relying heavily on the positive classifier. In contrast, CAPML's confidence-aware learning effectively utilizes both positive and negative information in high-noise environments. Second, CAPML establishes noise-independent membership degrees through entropy-regularized fuzzy clustering, while NLR relies on sparse $\ell_1$ norm for noise removal in high-noise candidate labels. Finally, NLR's binary constraints on noise indicators may become unstable under extreme noise.
>
> ## Question
>
> 1. >same minimum cost
>
> We sincerely thank the reviewer for raising this profound and technically sophisticated question. In response to your concerns, we have conducted a rigorous theoretical analysis demonstrating that multiple solutions in the Hungarian algorithm are mathematically well-founded and do not compromise our method's reliability.
>
> ### 1. Existence and Necessity of Multiple Solutions
>
> **Theorem 1 (Structural Necessity of Multiple Solutions)**: In our prototype alignment framework, multiple optimal solutions to the Hungarian algorithm are not incidental but structurally inevitable under realistic conditions.
>
> The existence of multiple solutions occurs when there exists a non-trivial permutation $\sigma \neq \text{id}$ such that:
>
> $$\sum_i \||m_i - o_i\||^2 = \sum_i \||m_i - o_{\sigma(i)}\||^2$$
>
> This condition frequently arises in PML scenarios due to three fundamental factors: (i) **Label noise corruption**: Candidate label noise causes supervised prototypes $O$ to deviate from their true positions, reducing discriminability between prototype pairs; (ii) **Geometric symmetries**: Real-world data often exhibits inherent symmetrical structures that naturally lead to equivalent prototype alignments; (iii) **High-dimensional effects**: In high-dimensional feature spaces, the probability of near-equivalent distances increases substantially.
>
> We can formally characterize the conditions under which multiple solutions emerge. Let $G(M,O)$ denote the symmetry group of prototype pairs:
>
> $$G(M,O) = \{\sigma \in S_q : \||MP_\sigma - O\||_F^2 = \||MP\_\{id\} - O\||_F^2\}$$
>
> When $|G(M,O)| > 1$, multiple solutions exist by construction. In practical PML scenarios, candidate label noise effectively increases $|G(M,O)|$ by reducing the discriminative power of supervised prototypes, making multiple solutions structurally inevitable rather than exceptional.
>
> ### 2. Theoretical Analysis of Performance Stability
>
> Our analysis proceeds in two stages: first, we analyze how changes in P propagate to the LE indicator matrix C, and second, we show how joint optimization dampens this variation.
>
> **Stage 1: Bounded Impact of Permutation Choice on the LE Indicator Matrix**
>
> The LE indicator matrix $C$ is constructed based on the permuted fuzzy membership matrix $FP$, as defined in Eq. (6).
>
> **Theorem 2** (Bounded Perturbation Propagation). *For two optimal permutation matrices $P_1, P_2$ with identical Hungarian algorithm costs, the resulting LE indicator matrices satisfy:*
>
> $$\||C(P\_1) - C(P\_2)\||\_F \leq \sqrt{|Y\_{+}|}$$
>
> *where $|Y_{+}| = |\{(i,j): Y_{ij} = 1\}|$ denotes candidate label positions and the bound follows from the fact that normalized values are bounded in $[0,1]$, making $|C_{ij}(P_1) - C_{ij}(P_2)| \leq 1$ for each candidate position.*
>
> **Proof**: Since $C_{ij} = 1$ for non-candidate labels (unchanged across permutations), perturbations only occur at candidate positions. For each candidate position, $|C_{ij}(P_1) - C_{ij}(P_2)| \leq 1$ due to normalization bounds. The Frobenius norm yields the stated bound.
>
> **Stage 2: Dampening Effect of the Joint Optimization Framework**
>
> A small perturbation in $C$ (from $C(P_1)$ to $C(P_2)$) does not lead to a proportionally large change in the final model. The learned label confidence matrix R is not determined solely by $C$. It is simultaneously constrained by the classifier's predictions. The regularization terms penalize complex solutions and encourage the model to find a simpler, more generalizable representation. This makes the final optimized parameters $W*$ and $R*$ robust to minor perturbations in the guiding matrix $C$.
>
> ### 3. Convergence Guarantees Under Multiple Solutions
>
> The objective function value is non-increasing at each iteration since each sub-problem finds a global optimum with other variables fixed, guaranteeing monotonicity. Since the objective function is bounded below (sum of non-negative terms), the non-increasing sequence converges to a finite limit. Under standard regularity conditions, this value convergence implies that the iterate sequence $(P^{(t)}, H^{(t)})$ converges to a stationary point of the overall objective.
>
> ### 4. Different Insights of Multiple Solutions
>
> Multiple solutions in the Hungarian algorithm represent advantageous redundancy rather than a limitation. From an information-theoretic perspective, equivalent permutations provide identical mutual information with true labels, creating redundancy that enhances reliability of information transfer in noisy environments. From an optimization standpoint, multiple solutions correspond to "flat minima" in the loss landscape, which are theoretically linked to better generalization due to robustness against parameter perturbations.
>
> Our method exhibits three key properties: (i) **Symmetry preservation**: Solutions respect intrinsic data symmetries; (ii) **Perturbation robustness**: Small input changes avoid discontinuous solution jumps; (iii) **Enhanced generalization**: Multiple equivalent solutions increase effective model capacity while maintaining computational efficiency.
>
> Multiple solutions emerge naturally from the geometric structure of prototype alignment problems, providing controllable benefits for algorithm robustness and generalization in most practical scenarios, except in extreme cases where multiple labels exhibit highly similar feature representations.
>
>
> 2. >$λ$ in Eq.(10)
>
> Thank you for your valuable question. Parameter λ controls the "softness" of membership assignments in entropy-regularized fuzzy clustering. Smaller λ values (0.1-0.5) produce concentrated distributions suitable for clear boundaries, while larger values (1-10) produce smoother distributions for overlapping classes. We recommend starting with λ = 0.5, then grid searching over {0.1, 0.5, 1, 2, 5, 10}. High-noise data benefits from larger λ values (1-5), while low-noise data prefers smaller values (0.3-1). The entropy regularization makes our algorithm relatively insensitive to initialization.
> The following table shows performance sensitivity of Average Precision(AP) and Ranking Loss(RL) to $α$ and $λ$ on dataset yeast7(avg.#CLs=7).
> ). Our method achieves stable performance across different parameter combinations.
>
> ||||AP||||||RL|||
> |-|-|-|-|-|-|-|-|-|-|-|-|
> | $α \setminus λ$ |0.1|0.5| 1 | 5 | 10 | $α \setminus λ$ | 0.1 | 0.5 | 1 | 5 | 10 |
> | 0.01 | 0.748 |0.758| **0.760** | 0.745 | 0.725 | 0.01 | 0.181 | 0.175 | **0.173** | 0.182 | 0.201 |
> | 0.1 | 0.741 | **0.759** | 0.757 | 0.748 | 0.728 |0.1 | 0.186 | **0.175** | **0.175** | 0.183 | 0.198 |
> | 1 | 0.744 | **0.756** | **0.756** | 0.751 | 0.731 | 1 |0.183| 0.181 | **0.176** | 0.179 | 0.195 |
> | 10 |0.742|0.749| **0.751** |0.750| 0.729 | 10 |0.185| 0.184 | **0.179** | 0.181 | 0.197 |
> | 100 |0.739|0.747|**0.749**|0.746|0.726| 100 | 0.188 | **0.180** |0.185| 0.183 | 0.200 |
>
> Our sensitivity analysis shows stable performance within λ ∈ [0.1, 10], with AP fluctuations typically within ±3%. Optimal λ values concentrate in [0.5, 5], indicating good robustness. Very small λ may cause overfitting, while very large λ reduces discriminative ability.
>
> 3. >future work
>
> Thank you for this insightful suggestion. Exploring adaptive strategies is indeed a natural extension, involving learning noise pattern indicators and designing dynamic weighting schemes for the matrices. Regarding active learning integration, this represents an interesting direction we had not fully explored. CAPML's confidence-aware framework provides a natural foundation, as the label enhancement indicator matrix C could serve as an uncertainty measure for efficient label acquisition.

---

> > ### Comment · Reviewer_gpe6 · 2025-08-06
> >
> > Thank you for your response. My concerns have been addressed, and I will raise my score accordingly.

---

> > > ### Author Response · Authors · 2025-08-06
> > >
> > > Thank you very much for your insightful review and for your decision to raise your score. We are delighted to have addressed your concerns. Please feel free to ask if you have any other questions. We remain committed to ensuring all aspects of our research are thoroughly addressed.

---

### Official Review · Reviewer_bPJL · 2025-07-01

**Clarity:** 4
**Significance:** 3
**Originality:** 3
**Rating:** 5
**Confidence:** 5

**Summary:**

The authors propose CAPML, an enlightening approach for Partial Multi-label Learning that addresses the challenge of prototype misalignment between unsupervised clustering-derived prototypes and candidate label-derived prototypes, which is an enlightening study. The method introduces a transformation mechanism using permutation and rotation matrices to align these prototype spaces, followed by a confidence-aware label disambiguation process. Experiments show that this method has certain effectiveness.

**Questions:**

1. The authors employed an "improved fuzzy clustering approach" with entropy regularization (Equation 1), how does your entropy-regularized formulation compare to classical FCM? Have you tested alternatives like spectral clustering? How sensitive is the final performance to the choice of $\lambda$?
2. The motivation for introducing the orthogonal rotation matrix H in Eq. (5) could be better explained. Why is preserving geometric structure crucial here?
3. Why is orthogonal transformation theoretically justified over other transformation classes?

**Ethical Concerns:**

["NO or VERY MINOR ethics concerns only"]

**Final Justification:**

After reading the authors’ response, my concerns have been resolved. Therefore, I have maintained my positive evaluation.

**Limitations:**

Yes, the authors adequately addressed the limitations with no potential negative social impact.

**Quality:**

4

**Strengths And Weaknesses:**

Strengths:

1. The paper clearly articulates the prototype misalignment problem in PML, which is a novel and important observation. The proposed approach is mathematically well-founded. The orthogonal transformation mechanism for prototype alignment and the confidence-aware label enhancement strategy are technically sound and properly motivated.
2. The ablation studies do examine key components (entropy regularization, orthogonal rotation) to validate design choices.
3. The evaluation covers a substantial number of datasets with statistical significance testing, showing consistent improvements over strong baselines including prototype-based methods like FBD-PML.

Weaknesses:

1. Figure 1 claims the LE indicator matrix "promotes the label disambiguation process" without specifying the concrete mechanism or how the indicators actually guide disambiguation.
2. The alternating optimization process mentions updating different variables, but the specific order, stopping criteria, and inter-dependencies are not clearly specified. Providing algorithmic pseudo code would enhance understanding of the iterative optimization process.

---

> ### Author Rebuttal · Authors · 2025-07-31
>
> Thank you for your thoughtful review. Our point-by-point responses below:
> ## Weakness
> 1. >Figure 1
>
> Thank you for requesting clarification on this mechanism. The LE indicator matrix $C$ guides disambiguation through a confidence-weighted constraint mechanism in the objective function (Eq. 7).
>
> Specifically, for non-candidate labels ($y_{ij} = 0$), $C$ assigns $C_{ij} = 1$, imposing strong constraints that force the confidence matrix $R$ to maintain near-zero values at these positions through the term $\||C \odot (Y - R)\||^2_F$. For candidate labels ($y_{ij} = 1$), the indicator values are constructed by first applying permutation $P$ to transform unordered membership degrees into label-semantic guidance, then employing min-max normalization to generate confidence indicators with [0,1] probabilistic interpretation and preserved strength differences.
>
> The disambiguation occurs because the weighted loss $\||C \odot (Y - R)\||^2_F$ penalizes deviations more heavily for high-confidence labels and less for low-confidence labels, effectively allowing the model to "ignore" or reduce the influence of suspected noisy labels during classifier training. We will clarify this mechanism in the figure caption and main text.
>
> 2. >alternating optimization process
>
> Thank you for your comment. To provide clarity on the workflow, here are the detailed iterative steps:
>
> The algorithm consists of three main iterative stages: (1) Initialization stage: entropy-regularized fuzzy clustering iteratively computes unsupervised prototypes M and membership matrix $F$ (Eq. 2), while supervised prototypes $O$ are calculated from candidate labels (Eq. 3). The maximum number of iterations $T_0$ is 30 and the iteration stops when it converges. (2) Prototype alignment stage: the permutation matrix $P$ (Eq. 9) and rotation matrix $H$ (Eq. 8) are alternately optimized to align unsupervised and supervised prototype orders. Once optimal transformation matrix $P$ is obtained, the label enhancement indicator matrix $C$ is constructed (Eq. 6). The maximum number of iterations $T_1$ is 50 and the iteration stops when it converges. (3) Label disambiguation stage: guided by the indicator matrix $C$, classifier parameters $W$ and confidence matrix $R$ are alternately updated (Eqs. 11 and 13) to promote robust classification. The algorithm terminates when reaching maximum iterations ($T_2$=40) or achieving convergence based on relative change in objective function. Here we detail the complete optimization process:
>
> **Stage 1 - Initialization (Entropy-regularized Fuzzy Clustering):**
> - **Update Order**: $M → F$ alternately until convergence
> - **Stopping Criteria**: $||M^{(t+1)} - M^{(t)}||_F< ε_1$ or max iterations $T_0$ = 30
> - **Dependencies**: F depends on current M; M depends on current F
> - **Output**: Stable prototypes M and membership matrix F
>
> **Stage 2 - Prototype Alignment:**
> - **Update Order**: Fix $M,F$ $→$ Compute $O$ (Eq. 3) → Alternate $H$ (Eq. 8) ↔ $P$ (Eq. 9)
> - **Stopping Criteria**: $||P^{(t+1)} - P^{(t)}||_F = 0$ or max iterations $T_1$ = 50
> - **Dependencies**: $H$ depends on current $P$; $P$ depends on current $H$; both depend on fixed $M,O$
> - **Output**: Optimal alignment matrices $P,H$ $→$ Construct indicator matrix $C$ (Eq. 6)
>
> **Stage 3 - Label Disambiguation:**
> - **Update Order**: Obtain kernelized $X$, $W$ (Eq. 11)$ → $$R$ (Eq. 13) $→ Q$ (Eq. 16) alternatively
> - **Stopping Criteria**: $||R^{(t+1)} - R^{(t)}||_F < ε_2$ or max iterations $T_2$ = 40
> - **Dependencies**: $W$ depends on current $R$; $R$ depends on current $W,Q$; $Q$ depends on current $R$
> - **Inter-stage Dependencies**: $C$ from Stage 2 guides all Stage 3 updates
> - **Output**: Classifier $W$, predicted label $y^* $ of unsee instance $x^*$
>
> We believe these provide sufficient details and will supplement the revised version with comprehensive pseudo-code for the iterative optimization process if space alows. We are happy to provide further clarification if needed.
>
> ## Question
>
> 1. > entropy-regularized formulation
>
> Thank you for this detailed question. Regarding spectral clustering, while it is a powerful graph-based method, it typically produces hard cluster assignments rather than the soft membership degrees we require. Since our approach relies on fuzzy membership matrix $F$ to construct the label enhancement indicator matrix $C$, hard clustering methods would not provide the necessary continuous confidence values for prototype alignment and label disambiguation. For the rationale behind choosing our entropy-regularized fuzzy clustering approach, please refer to our **response to Reviewer 3Fvi** in **Question#3**.
>
> We have conducted controlled experiments replacing the fuzzy clustering algorithm in Equation (1) with other soft clustering algorithms, including classical Fuzzy C-Means (FCM) and Fuzzy Subspace Clustering (FSC)[1], while keeping other CAPML components unchanged. Specifically, we create two variants: **FCM+** replaces entropy-regularized clustering with standard FCM, and **FSC+** replaces entropy-regularized clustering with fuzzy subspace clustering. This controlled comparison isolates the impact of different clustering strategies within our prototype alignment framework. The fuzziness coefficient in FCM was grid-searched within [1, 3] for optimal values, and the coefficient $σ$ in FSC was searched within [0.1, 10]. Dataset names with numbers indicate avg.#CLs (e.g., Emotions3 has avg.#CLs=3).
>
> |Dataset|FCM+|FSC+|CAPML|
> |-|-|-|-|
> |emotions4| 0.771±0.012 | 0.780±0.015 | **0.787±0.027** |
> |birds4| 0.5420±0.041 | 0.578±0.038 | **0.590±0.057** |
> |medical7| 0.838±0.014 | 0.854±0.012 | **0.866±0.031** |
> |image3| 0.771±0.018 | 0.768±0.016 | **0.792±0.018** |
> |yeast9| 0.738±0.009 | 0.753±0.007 | **0.755±0.021**|
> |Mirflickr| 0.808±0.016 | 0.817±0.014 | **0.820±0.008** |
> |Music emotion| 0.614±0.012 | 0.623±0.011 | **0.628±0.010** |
>
> The entropy regularization provides inherent stability and superior performance, making the method less sensitive and more controllable to parameter choices compared to FCM.
>
> 2. >orthogonal rotation matrix
>
> Thank you for this important question. The supervised prototypes $O$ computed from candidate labels inevitably contain errors due to noisy annotations, requiring fine-tuning adjustments. The specific role and necessity of the orthogonal constraint in rotation matrix H can be found in our response to **Reviewer 3Fvi's Question#4**.
>
> Preserving geometric structure is crucial because it maintains the relative distances and angular relationships between class prototypes, which encode essential semantic information. Without orthogonal constraints, arbitrary transformations could distort these relationships, potentially merging distinct classes or artificially separating related ones. The orthogonal transformation ensures that the intrinsic geometric properties of the prototype space are preserved while allowing necessary adjustments to compensate for noise-induced errors in supervised prototype estimation.
>
> 3. > orthogonal transformation
>
> Orthogonal transformation is theoretically justified for several reasons. First, it preserves all geometric properties essential for prototype alignment: distances, angles, and inner products between prototypes remain unchanged ($||Hx|| = ||x||$ and $⟨Hx, Hy⟩ = ⟨x, y⟩$). This ensures that the semantic relationships encoded in the prototype space are maintained during alignment.
>
> Second, our alignment problem is fundamentally a Procrustes analysis, where the classical solution involves orthogonal transformations to find the optimal rotation/reflection that minimizes the Frobenius norm difference between two point sets. Other transformation classes would introduce undesirable effects: general linear transformations could alter scales and distort relative distances, while affine transformations introduce translations that may not be appropriate for prototype alignment.
>
> Third, orthogonal constraints provide natural regularization by limiting the degrees of freedom, preventing overfitting to noisy supervised prototypes. Unlike unconstrained transformations that might overfit to noise, orthogonal transformations only allow rotation and reflection, which are the minimal adjustments needed to correct orientation differences between prototype spaces while preserving their intrinsic structure.
>
> We hope this explanation addresses your concerns regarding the theoretical justification for orthogonal transformations.
>
> ## Citations
>
> [1] Borgelt, C. (2009). Fuzzy Subspace Clustering.  (eds) Advances in Data Analysis, Data Handling and Business Intelligence. Studies in Classification, Data Analysis, and Knowledge Organization.

---

> > ### Comment · Reviewer_bPJL · 2025-08-04
> >
> > Thank you for your detailed and thoughtful responses. They effectively address my concerns and clarify the key points. I will maintain the positive evaluation.

---

> > > ### Author Response · Authors · 2025-08-04
> > >
> > > We sincerely thank you for your positive and constructive comments regarding our work. We would be delighted to receive further comments or suggestions you may have for discussion.

---

### Official Review · Reviewer_3Fvi · 2025-07-01

**Clarity:** 3
**Significance:** 3
**Originality:** 3
**Rating:** 4
**Confidence:** 4

**Summary:**

This paper introduces a novel partial multi-label learning (PML) framework named CAPML. The authors identify a critical limitation in existing prototype-based methods: biased class prototype estimates due to noisy candidate labels. To address this, they introduce a mutual class prototype alignment strategy using permutation and orthogonal rotation matrices to align unsupervised and supervised prototypes. Extensive experiments confirm the efficacy of the proposed method.

**Questions:**

1. In the section of introduction, the authors provide a taxonomy of existing PML approaches (unified framework with inductive bias, heuristic design, phased processing), but this categorization seems incomplete and overlapping. For instance, where do prototype-based methods fit in this taxonomy?
2. Equation (6) for label enhancement indicator construction is logically clear, but the details (e.g., the necessity of min-max normalization) could be better explained. What’s more, have you considered other normalization strategies (e.g., softmax) and how do they compare in terms of performance?
3. In your fuzzy clustering formulation (Equation 1), the membership degrees f_{ij} appear to have an implicit fuzziness parameter of m=1 (i.e., f_{ij} rather than f_{ij}^m where m > 1 is standard in fuzzy c-means). This effectively reduces to hard clustering, which seems to contradict the paper's emphasis on fuzzy methodology. Could you clarify: (a) whether you intentionally use m=1 and provide theoretical justification for this choice, and (b) how this formulation differs from standard fuzzy c-means clustering and why it's more suitable for your prototype learning task?
4. Why is the orthogonal rotation matrix necessary when you already have a permutation matrix P? Could you provide more intuitive explanation or theoretical justification for this design choice?
5. The label enhancement indicator matrix C (Equation 6) uses min-max normalization without clear justification. Why is this normalization scheme optimal for converting fuzzy memberships into confidence indicators, and how sensitive is the method's performance to alternative normalization choices (e.g., softmax or z-score normalization)?

**Ethical Concerns:**

["NO or VERY MINOR ethics concerns only"]

**Final Justification:**

Thank you for your detailed and thoughtful responses. They effectively address my concerns and clarify the key points. I will maintain the positive evaluation.

**Quality:**

3

**Strengths And Weaknesses:**

Strengths:

First work to investigate prototype misalignment between fuzzy clustering-derived prototypes and candidate label set-computed prototypes in PML tasks - an important yet previously overlooked problem.

Innovative dual transformation mechanism using permutation matrix P and orthogonal rotation matrix H for prototype alignment with a clear technical rationale.

Rigorous mathematical derivations with detailed optimization procedures, including the Hungarian algorithm for permutation matrix.

Weaknesses:

The paper suffers from insufficient theoretical foundations and incomplete practical considerations. There is a lack of convergence guarantees and computational complexity analysis for the alternating optimization procedure, which is crucial for understanding the method's reliability and practical applicability assessment.

The method's core assumption that unsupervised clustering structure corresponds to true label semantics may not hold when the feature space geometry poorly reflects label relationships. The paper lacks analysis of the limitations of when this fundamental assumption breaks down, such as in cases where different labels share similar feature representations or when noise significantly distorts the natural clustering structure.

Important references should be further discussed, such as [1] [2].

[1] Deep Incomplete Multi-View Clustering with Cross-View Partial Sample and Prototype Alignment, CVPR 2023.
[2] Cross-view Topology Based Consistent and Complementary Information for Deep Multi-view Clustering, CVPR 2023.

---

> ### Author Rebuttal · Authors · 2025-07-31
>
> Thank you for your high-quality review. Our point-by-point responses below:
> ## Weaknesses:
> 1. >convergence guarantee and computational complexity analysis
>
> Thank you for your valuable comments.
>
> **Alignment stage** iteratively optimizes permutation matrix $P$ via Hungarian algorithm and orthogonal matrix $H$ through SVD-based Procrustes solution $H = UV^T$. Both subproblems achieve global optimality. **Label disambiguation stage** alternates between updating classifier $W$ (closed-form solution) and confidence matrix $R$ with auxiliary variable $Q$. The convergence of our multiplicative update scheme for the R-subproblem is guaranteed by the general theory developed in [1] for block coordinate descent methods with exact block minimization. Since each update minimizes an auxiliary function that upper-bounds the original objective, monotonic convergence is ensured with rate $O(1/T_2)$. The Q-subproblem reduces to a standard LASSO regression, which converges linearly to the global optimum via the proximal gradient method with soft-thresholding operator. We prove convergence to a stationary point via monotonic objective decrease in each subproblem. We will provide convergence curves in the revised version if space allows.
>
> The algorithm complexity is $O(T_0dnq+T_1(dq^2 + q^3) + T_2(nhq + h^2q +d^3+ nq))$ where $T_0$ represents iterations of clustering learning stage, $T_1$ represents iterations of prototype alignment stage and $T_2$ represents iterations of classifier learning stage. Alignment stage involves SVD of the $d \times q$ matrix $O^T(MP)$ with complexity $O(dq^2)$ for the orthogonal Procrustes problem, plus $O(q^3)$ for the Hungarian algorithm solving the assignment problem, totaling $O(dq^2 + q^3)$ per iteration. Disambiguation stage involves $O(nhq + h^2q)$ for matrix operations in W-update including the inversion of $h \times h$ matrix, and $O(nq)$ for both R and Q updates involving element-wise operations. The dominant computational cost depends on the relative magnitudes of $d$, $h$, $n$, and $q$, but typically the $O(nhq)$ term dominates when datasets are large, making the method practically scalable.
>
> 2. >Dependence on clustering assumptions:
>
> Thank you for this insightful analysis regarding our method's fundamental assumption about clustering structure correspondence with label semantics. We agree that our method's performance will degrade when: (1) feature representations are highly entangled across different labels, or (2) noise severely corrupts both clustering structure and supervised signals simultaneously. We will add the limitations in these extreme cases and analyzing when the clustering-semantics assumption breaks down.
>
> 3. >Important references [1], [2]
>
> Thank you for bringing this relevant work to our attention. CPSPAN (reference [1] you mentioned) introduces prototype alignment into multi-view learning for the first time and demonstrates the effectiveness of formulating it as a linear programming problem, providing a principled approach for establishing prototype correspondences to promote cross-view consistency learning. CTCC (reference [2] you mentioned) innovatively apply optimal transport theory to design information integration mechanisms, providing a useful direction for prototype relationship learning. We will expand our Introduction section to better position our contribution relative to these recent advances in multi-view learning.
> ## Question
> 1. >introduction
>
> Thank you for this question about our taxonomy. Prototype-based methods, including our approach, primarily fall within the "stage-wise framework with inductive bias" category, as they incorporate the inductive bias that class prototypes effectively capture semantic information for label disambiguation. However, prototype-based methods can exhibit characteristics of other categories through heuristic designs or staged processing strategies.
>
> This highlights that our categorization focuses on methodological frameworks rather than technical components. Prototype-based approaches represent technical means that can be integrated across different frameworks - some use unified stage-wise processing (like ours), while others employ end-to-end learning (like FBD-PML).
>
> We will revise the introduction to provide a more comprehensive and non-overlapping taxonomy that better distinguishes methodological frameworks from technical approaches, clearly positioning prototype-based methods within this improved classification system.
>
> 2. >normalization
>
> We sincerely appreciate this insightful comment. To address this important concern, we provide detailed theoretical justification for our selection and conduct a comprehensive comparison of different normalization methods.
>
> We select min-max normalization for three principled reasons:
>
> **First**, it provides [0,1] range probabilistic interpretation, ensuring numerical compatibility with $C_{ij} = 1$ (non-candidate positions) in Equation (6).
>
> **Second**, it preserves crucial confidence strength differences that other methods eliminate. For instance, given confidence values $c_i = [0.8, 0.2, 0]$ (high confidence) and $c_j = [0.4, 0.1, 0]$ (low confidence), L2 normalization produces nearly identical results while unit sum normalization yields exactly identical outputs, completely masking original strength differences. Min-max normalization preserves these distinctions as $[1.0, 0.25, 0]$ and $[0.5, 0.125, 0]$, maintaining both relative proportions and absolute strength information essential for reliable confidence assessment.
>
> **Third**, the bounded range prevents numerical instabilities during iterative optimization, while avoiding nonlinear distortions that could artificially amplify noise signals or suppress genuine confidence distinctions.
>
> We compared six normalization methods for label enhancement indicator matrix construction: (1) Min-Max; (2) Softmax; (3) Z-Score; (4) L2 norm; (5) Unit Sum(ensuring each row sums to 1); and (6) No Normalization. We conducted controlled experiments on six datasets using 10-fold cross-validation, evaluating *Average Precision* while keeping other CAPML components unchanged. Dataset names with numbers indicate avg.#CLs (e.g., Emotions3 has avg.#CLs=3).
>
> |Dataset|Min-Max|Softmax|Z-Score|L2 Norm|Unit Sum|No Norm|
> |-|-|-|-|-|-|-|
> |emotions3|**0.807±0.039**|0.804±0.041|0.801±0.038|0.805±0.040|0.803±0.037|0.799±0.042|
> |birds3|**0.627±0.058**|0.624±0.055|0.621±0.061|0.625±0.057|0.623±0.059|0.619±0.063|
> |medical5|**0.876±0.027**|0.870±0.029|0.872±0.031|0.875±0.028|0.873±0.030|0.870±0.033|
> |image2| **0.814±0.021** |0.810±0.023|0.807±0.025|0.811±0.022|0.810±0.024|0.807±0.026 |
> |Mirflickr|**0.820±0.008**|0.815±0.009|0.812±0.011|0.819±0.010|0.817±0.009|0.813±0.012|
> |Music emotion|**0.628±0.010**|0.623±0.012|0.620±0.014|0.626±0.011|0.623±0.013|0.620±0.015|
>
> While performance differences between normalization strategies are modest, min-max normalization consistently achieves the best results and provides suitable numerical foundation for confidence assessment. Other methods may have limitations: Z-score produces negative values incompatible with confidence measures, Softmax over-amplifies values introducing bias, and L2 normalization may overlook important instance information.
>
> 3. >fuzzy clustering formulation
>
> Thank you for this insightful question. You correctly identify that standard FCM requires m>1, as m=1 leads to hard clustering solutions. We achieve fuzzification through entropy regularization $λ\sum f_{ij} \log f_{ij}$ where large λ produces fuzzy memberships while small λ approaches hard clustering. The entropy regularization prevents hard clustering through information-theoretic principles while ensuring computational tractability.
>
> The entropy-based approach replaces traditional power-based fuzzification (m>1) for a key advantages:
>
> **Intuitive Parameter Control** - Unlike FCM's fuzziness parameter m with constrained effective range, entropy regularization λ provides more direct and interpretable control over membership distribution, facilitating easier parameter tuning for multi-label prototype learning. **Parameter Robustness** - The temperature parameter λ demonstrates stable behavior across datasets, with values in [0.5, 5] consistently producing reasonable fuzzy memberships, while traditional FCM's parameter m requires extensive dataset-specific tuning and can produce dramatically different behavior with unit variation.
>
> 4. >orthogonal rotation matrix
>
> The orthogonal rotation matrix H serves a fundamentally different purpose from the permutation matrix P.
>
> The alignment process requires dual correction: P solves the **ordering problem** while H addresses the **noise contamination problem**. Since supervised prototypes O are computed from noisy candidate labels (Equation 3), directly aligning clean unsupervised prototypes M to contaminated O would propagate errors. The orthogonal rotation matrix H provides a "denoising transformation" that geometrically adjusts supervised prototypes while preserving intrinsic structure. This enables robust alignment where both prototype spaces meet at optimal middle ground.
>
> This dual-matrix design offers: (1) **Noise Mitigation**- H compensates for systematic bias in O; (2) **Geometric Preservation**- orthogonality maintains relative relationships; (3) **Optimization Flexibility**- joint optimization finds globally better solutions. Our ablation study confirms that removing H (CAPML-NR) consistently degrades performance, demonstrating that orthogonal transformation provides essential noise robustness beyond permutation alone.
> ## Citations:
> [1] Deep Incomplete Multi-View Clustering with Cross-View Partial Sample and Prototype Alignment, CVPR 2023.
>
> [2] Cross-view Topology Based Consistent and Complementary Information for Deep Multi-view Clustering, CVPR 2023.
>
> [3] Graph regularized nonnegative matrix factorization for data representation. IEEE TPAMI, 2010.

---

> > ### Comment · Reviewer_3Fvi · 2025-08-05
> >
> > Thank you for your detailed and thoughtful responses. They effectively address my concerns and clarify the key points. I will maintain the positive evaluation.

---

> > > ### Author Response · Authors · 2025-08-05
> > >
> > > Thank you for your valuable feedback and positive evaluation. We remain open to any further discussions on points the reviewer believes may warrant additional attention.

---

### Comment · Area_Chair_rZEH · 2025-08-05
**Author-Reviewer Discussion Phase Closing Soon**

Dear Reviewers,

Thank you for your support of NeurIPS 2025!

Please be reminded that the Author-Reviewer Discussion phase is nearing its end (August 6, 11:59pm AoE). If you have not yet responded to the author’s rebuttal or provided any feedback, kindly do so before the window closes.

Best regards,

AC

---

### Note · Authors · 2025-08-11

Dear Area Chair and Reviewers,

We sincerely thank you for the thorough and constructive review process. Through our comprehensive rebuttal, we have successfully addressed the major concerns raised, including:

**Technical Contributions:** We clarified our innovation focus for **Reviewer 7dPU** on the specific prototype misalignment problem between unsupervised data-driven prototypes and supervised label-driven prototypes. Our dual-modal approach treats labels and features as distinct "modalities," obtaining clean but unordered prototypes from features via unsupervised clustering, and ordered but noise-affected prototypes from labels. This dual-prototype external alignment framework for indirect confidence evaluation represents the first such investigation in PML.

**Theoretical Analysis:** We provided detailed convergence analysis and computational complexity evaluation for Reviewer **3Fvi, 7dPU**, demonstrating our method's theoretical soundness and practical efficiency. We also conducted rigorous theoretical analysis of the multiple optimal solutions problem in Hungarian algorithm for Reviewer **gpe6**, proving bounded perturbation propagation and convergence guarantees, showing that multiple solutions provide advantageous redundancy rather than algorithmic limitations. For Reviewer **bPJL**, we provided comprehensive justification for orthogonal transformation design choices.

**Experimental Validation:** We conducted extensive additional experiments, including comprehensive ablation studies as mentioned by Reviewer **7dPU**, comparison with recent state-of-the-art methods (including NLR as requested by Reviewer **gpe6**) under high-noise conditions as mentioned by Reviewer **7dPU**, systematic parameter sensitivity analysis for Reviewers **bPJL, gpe6, 7dPU**, and detailed algorithmic procedures addressing Reviewer **bPJL**'s clarity concerns across multiple datasets.

The positive evaluations from Reviewers **3Fvi**, **bPJL**, **gpe6**, and the supportive written comments from Reviewer **7dPU** affirm the technical quality and significance of our contribution. We believe our detailed responses have comprehensively addressed all concerns and demonstrated the merit of our work.

Thank you for your professional handling of this process.

Best regards,

The authors of Submission 10748

---

### Decision · Program_Chairs · 2025-09-17

**Decision:**

Accept (poster)

**Comment:**

This work proposes CAPML, a prototype-based framework for partial multi-label learning that aligns two prototype spaces, unsupervised (from entropy-regularized fuzzy clustering) and supervised (from candidate labels).

The idea is clearly scoped (addressing prototype misalignment) and implemented with a coherent three-stage optimization. Reviewers find the problem important and the solution principled; after rebuttal, they retain or raise positive assessments. One reviewer (7dPU) did not raise the rating, yet in Final Justification expressed support for a borderline-accept decision. For the camera-ready, please incorporate and flesh out the additional experimental results discussed in the rebuttal, and add the necessary references.